# ON THE EQUIVALENCE BETWEEN POSITIONAL NODE EMBEDDINGS AND STRUCTURAL GRAPH REPRESENTATIONS

**Balasubramaniam Srinivasan**
Department of Computer Science
Purdue University
bsriniv@purdue.edu

**Bruno Ribeiro**
Department of Computer Science
Purdue University
ribeiro@cs.purdue.edu

## ABSTRACT

This work provides the first unifying theoretical framework for node (positional) embeddings and structural graph representations, bridging methods like matrix factorization and graph neural networks. Using invariant theory, we show that the relationship between structural representations and node embeddings is analogous to that of a distribution and its samples. We prove that all tasks that can be performed by node embeddings can also be performed by structural representations and vice-versa. We also show that the concept of transductive and inductive learning is unrelated to node embeddings and graph representations, clearing another source of confusion in the literature. Finally, we introduce new practical guidelines to generating and using node embeddings, which fixes significant shortcomings of standard operating procedures used today.

## 1 INTRODUCTION

The theory of *structural* graph representations is a recently emerging field. It creates a link between relational learning and invariant theory. Interestingly, or rather unfortunately, there is no unified theory connecting node embeddings —low-rank matrix approximations, factor analysis, latent semantic analysis, etc.— with structural graph representations. Instead, conflicting interpretations have manifested over the last few years, that further confound practitioners and researchers alike.

For instance, consider the direction, *word embeddings → structural representations*, where the structural equivalence between *men → king* and *women → queen* is described as being obtained by just adding or subtracting their node embeddings (positions in the embedding space) (Arora et al., 2016; Mikolov et al., 2013). *Hence, can all (positional) node embeddings provide structural relationships akin to word analogies?* We provide a visual example in Appendix (Section 7) using the food web of Figure 1. In the opposite direction, *structural representations → node embeddings*, graph neural networks (GNNs) are often optimized to predict edges even though their structural node representations are provably incapable of performing the task. For instance, the node representations of the *lynx* and the *orca* in Figure 1 are indistinguishable due to an isomorphic equivalence between the nodes, making any edge prediction task that distinguishes the edges of *lynx* and *orca* a seemly futile exercise (see Appendix (Section 7) for more details). *Hence, are structural representations in general —and GNNs in particular— fundamentally incapable of performing link (dyadic) and multi-ary (polyadic) predictions tasks?* GNNs, however, can perform node classification tasks, which is a task not associated with positional node embeddings (see Appendix (Section 7) for a concrete visual interpretation of the differences between positional node embeddings and structural representations over node classification and link prediction tasks).

Confirmation bias has seemingly appeared to thwart recent efforts to bring node embeddings and structural representations into a single overarching framework. Preconceived notions of the two being fundamentally different (see Appendix (Section 7)) have been reinforced in the existing literature, arguing they belong in different applications: (Positional) Node embeddings would find applications in multi-ary relationships such as link prediction, clustering, and natural language processing and knowledge acquisition through word and entity embeddings. Structural representations would find applications in node classification, graph classification, and role discovery. A unified theory is required if we wish to eliminate these artificial boundaries, and better cross-pollinate, node embeddings and structural representations in novel techniques.

*Contributions*: In this work we use invariant theory and axiomatic counterfactuals (causality) to develop a unified theoretical framework that clarifies the differences between node embeddings and

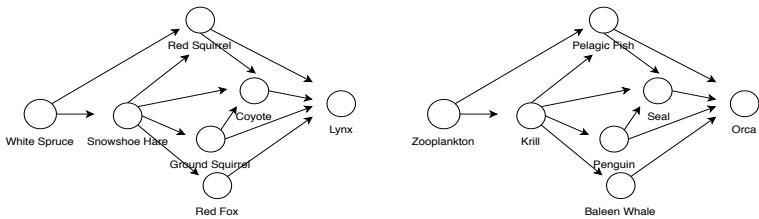

Figure 1: A food web example showing two disconnected components - the boreal forest (Stenseth et al., 1997) and the antarctic fauna (Bates et al., 2015). The positional node embedding of the lynx and the orca can be different while their structural representation must be the same (due to the isomorphism).

structural representations and emphasizes their correspondence. More specifically, (a) we show that structural representations and node embeddings have the same relationship as distributions and their samples; (b) we prove that all tasks that can be performed by node embeddings can also be performed by structural representations and vice-versa. Moreover, (c) we introduce new guidelines to creating and using node embeddings, which we hope will replace the less-than-optimal standard operating procedures used today. Finally, (d) we show that the concepts of *transductive* and *inductive* learning —commonly used to describe relational methods— are unrelated to node embeddings and structural representations.s

## 2 PRELIMINARIES

This section introduces some basic definitions, attempting to keep the mathematical jargon in check as much as we can, sometimes even sacrificing generality for clarity. We recommend Bloem-Reddy & Teh (2019) for a more formal description of some of the definitions in this section.

**Definition 1** (Graph). *We consider either a directed or an undirected attributed graph, denoted by $G = (V, E, \boldsymbol{X}, \mathbf{E})$, where $V$ is a set of $n = |V|$ vertices, $E$ is the set of edges in $V \times V$, with matrix $\boldsymbol{X} \in \mathbb{R}^{n \times k}, k > 0$ and 3-mode tensor $\mathbf{E} \in \mathbb{R}^{n \times n \times k'}, k' > 0$ representing the node and edge features, respectively. The edge set has an associated adjacency matrix $\boldsymbol{A} \in \{0, 1\}^{n \times n}$. In order to simplify notation, we will compress $\mathbf{E}$ and $\boldsymbol{A}$ into a single tensor $\mathbf{A} \in \mathbb{R}^{n \times n \times (k'+1)}$. When explicit vertex and edge features and weights are unavailable, we will consider $\boldsymbol{X} = \mathbf{1}\mathbf{1}^T$ and $\mathbf{A} = \boldsymbol{A}$, where $\mathbf{1}$ is a $n \times 1$ vector of ones. We will abuse notation and denote the graph as $G = (\mathbf{A}, \boldsymbol{X})$. Without loss of generality, we define number the nodes in $V = \{1, \ldots, n\}$ following the same ordering as the adjacency tensor $\mathbf{A}$ and the rows in $\boldsymbol{X}$. We denote $\vec{S}$ and a vector of the elements in $S \in \mathcal{P}^\star(V)$ sorted in ascending order, where $\mathcal{P}^\star(V)$ is the power set of $V$ without the empty set.*

One of the most important operators in our mathematical toolkit will be that of a permutation action, orbits, $\mathcal{G}$-invariance, and $\mathcal{G}$-equivariance:

**Definition 2** (Permutation action $\pi$). *A permutation action $\pi$ is a function that acts on any vector, matrix, or tensor defined over the nodes $V$, e.g., $(\boldsymbol{Z}_i)_{i \in V}$, and outputs an equivalent vector, matrix, or tensor with the order of the nodes permuted. We define $\Pi_n$ as the set of all $n!$ such permutation actions.*

**Definition 3** (Orbits). *An orbit is the result of a group action $\Pi_n$ acting on elements of a group correspond to bijective transformations of the space that preserve some structure of the space. The orbit of an element is the set of equivalent elements under action $\Pi_n$, i.e., $\Pi_n(x) = \{\pi(x) \mid \pi \in \Pi_n\}$.*

**Definition 4** ($\mathcal{G}$-equivariant and $\mathcal{G}$-invariant functions). *Let $\Sigma_n$ be the set of all possible attributed graphs $G$ of size $n \geq 1$. More formally, $\Sigma_n$ is the set of all tuples $(\mathbf{A}, \boldsymbol{X})$ with adjacency tensors $\mathbf{A}$ and corresponding node attributes $\boldsymbol{X}$ for $n$ nodes. A function $g : \Sigma_n \to \mathbb{R}^{n \times \cdot}$ is $\mathcal{G}$-equivariant w.r.t. valid permutations of the nodes $V$, whenever any permutation action $\pi \in \Pi_n$ in the $\Sigma_n$ space associated with the same permutation action of the nodes in the $\mathbb{R}^{n \times \cdot}$ space. A function $g : \Sigma_n \to \mathbb{R}^\cdot$ is $\mathcal{G}$-invariant whenever it is invariant to any permutation action $\pi \in \Pi_n$ in $\Sigma_n$.*

**Definition 5** (Graph orbits & graph isomorphism). *Let $G = (\mathbf{A}, \boldsymbol{X})$ be a graph with $n$ nodes, and let $\Pi_n(G) = \{(\mathbf{A}', \boldsymbol{X}') : (\mathbf{A}', \boldsymbol{X}') = (\pi(\mathbf{A}), \pi(\boldsymbol{X})), \forall \pi \in \Pi_n\}$ be the set of all equivalent (isomorphic) graphs under the permutation action $\pi$. Two graphs $G_1 = (\mathbf{A}_1, \boldsymbol{X}_1)$ and $G_2 = (\mathbf{A}_2, \boldsymbol{X}_2)$ are said isomorphic iff $\Pi_n(G_1) = \Pi_n(G_2)$.*

**Definition 6** (Node orbits & node isomorphism). *The equivalence classes of the vertices of a graph $G$ under the action of automorphisms are called vertex orbits. If two nodes are in the same node orbit, we say that they are isomorphic.*

In Figure 1, the lynx and the orca are isomorphic (they have the same node orbits). We now generalize Definition 6 to subsets of nodes $S \in \mathcal{P}^\star(V)$, where $\mathcal{P}^\star(V)$ is the power set of $V$ without the empty set.

**Definition 7** (Vertex subset orbits and joint isomorphism). *The equivalence classes of $k$-sized subsets of vertices $S \in \mathcal{P}^\star(V)$ of a graph $G$ under the action of automorphisms between the subsets are called vertex subset orbits, $k \geq 2$. If two proper subsets $S_1, S_2 \in \mathcal{P}^\star(V) \backslash V$ are in the same vertex subset orbit, we say they are jointly isomorphic.*

Next we define the relationship between structural representations and node embeddings.

# 3  A Unifying Theoretical Framework of Node Embeddings and Structural Representations

*How are node embeddings and structural representations related?* This section starts with a familiar, albeit naïve, view of the differences between node embeddings and structural representations, preparing the groundwork to later broadening and rectifying these into precise model-free mathematical statements using invariant theory. This broadening is needed since model-free node embeddings need not be related to *node closeness* in the graph (or to lower dimensional projections for that matter), as it is impossible to have a model-free definition of *closeness*.

*A familiar interpretation of node embeddings:* Node embeddings are often seen as a lower-dimensional projection of the rows and columns of the adjacency matrix $\boldsymbol{A}$ from $\mathbb{R}^n$ to $\mathbb{R}^d$, $d < n$, that preserves relative positions of the nodes in a graph (Graham & Winkler, 1985; Linial et al., 1995); for instance, in Figure 1, the lynx and the coyote would have close node embeddings, while the node embeddings of the lynx and the orca would be significantly different. Node embeddings are often seen as encoding the fact that the lynx and the coyote are part of a tightly-knit community, while the lynx and orca belong to distinct communities. The *structural representation* of a node, on the other hand, shows which nodes have similar roles (structural similarities) on a graph; for instance, the lynx and the orca in Figure 1 must have the same structural representation, while the lynx and the coyote likely have different structural representations. The lynx, like the orca, is a top predator in the food web while the coyote is not a top predator.

*The incompatibility of the familiar interpretation with the theory of structural graph representations:* The above interpretation of node embeddings must be tied to a model that defines *closeness*. Structural graph representations are model-free. Hence, we need a model-free definition of node embedding to connect it with structural representations. Unfortunately, one cannot define *closeness* without a model. Hence, in the remainder of this paper, we abandon this familiar interpretation in favor of a model-free definition.

*Roadmap:* In what follows, we restate some existing model-free definitions of *structural graph representation* and introduce some new ones. Then, we introduce a model-free definition of node embeddings. We will retain the terminology *node embedding* for historical reasons, even though our *node embedding* need not be an embedding (a projection into lower dimensional space).

## 3.1  On Structural Representations

In what follows we use the terms link and edge interchangeably. Proofs are left to the Appendix.

**Definition 8** (Structural node representations). *The structural representation of node $v \in V$ in a graph $G = (\mathbf{A}, \boldsymbol{X})$ is the $\mathcal{G}$-invariant representation $\Gamma(v, \mathbf{A}, \boldsymbol{X})$, where $\Gamma : V \times \Sigma_n \to \mathbb{R}^d$, $d \geq 1$, such that $\forall u \in V$, $\Gamma(u, \mathbf{A}, \boldsymbol{X}) = \Gamma(\pi(u), \pi(\mathbf{A}), \pi(\boldsymbol{X}))$ for all permutation actions $\forall \pi \in \Pi_n$. Moreover, for any two isomorphic nodes $u, v \in V$, $\Gamma(u, \mathbf{A}, \boldsymbol{X}) = \Gamma(v, \mathbf{A}, \boldsymbol{X})$.*

**Definition 9** (Most-expressive structural node representations $\Gamma^\star$). *A structural representation of a node $v \in V$, $\Gamma^\star(v, \mathbf{A}, \boldsymbol{X})$, is most-expressive iff, $\forall u \in V$, there exists a bijective measurable map between $\Gamma^\star(u, \mathbf{A}, \boldsymbol{X})$ and the orbit of node $u$ in $G = (\mathbf{A}, \boldsymbol{X})$ (Definition 7).*

Trivially, by Definitions 6 and 9, two graphs $G_1 = (\mathbf{A}_1, \boldsymbol{X}_1)$ and $G_2 = (\mathbf{A}_2, \boldsymbol{X}_2)$ are isomorphic (Definition 5) iff the most-expressive structural node representations $(\Gamma^\star(u, \mathbf{A}_1, \boldsymbol{X}_1))_{u \in V}$ and $(\Gamma^\star(v, \mathbf{A}_2, \boldsymbol{X}_2))_{v \in V}$ are the same up to a valid permutation $\pi \in \Pi_n$ of the nodes. In what follows $\mathcal{P}^\star$ is the power set excluding the empty set.

We now describe the relationship between structural node representations and node isomorphism.

**Lemma 1.** *Two nodes $v, u \in V$, have the same most-expressive structural representations $\Gamma^\star(v, \mathbf{A}, \boldsymbol{X}) = \Gamma^\star(u, \mathbf{A}, \boldsymbol{X})$ iff $u$ and $v$ are isomorphic nodes in $G = (\mathbf{A}, \boldsymbol{X})$.*

Having described representation of nodes, we now generalize these representations to subsets of $V$.

**Definition 10** (Joint structural representation $\Gamma$). *A joint structural representation of a graph with node set $V$ is defined as $\Gamma : \mathcal{P}^{\star}(V) \times \Sigma_n \to \mathbb{R}^d$, $d \geq 1$. Furthermore, $\Gamma$ is $\mathcal{G}$-invariant over all node subsets, i.e., $\forall S \in \mathcal{P}^{\star}(V)$ and $\forall (\mathbf{A}, \boldsymbol{X}) \in \Sigma_n$, it must be that $\Gamma(\vec{S}, \mathbf{A}, \boldsymbol{X}) = \Gamma(\pi(\vec{S}), \pi(\mathbf{A}), \pi(\boldsymbol{X}))$ for all permutation actions $\forall \pi \in \Pi_n$. Moreover, for any two isomorphic subsets $S, S' \in \mathcal{P}^{\star}(V)$, $\Gamma(\vec{S}, \mathbf{A}, \boldsymbol{X}) = \Gamma(\vec{S'}, \mathbf{A}, \boldsymbol{X})$.*

We now mirror Definition 9 in our generalization of $\Gamma$:

**Definition 11** (Most-expressive joint structural representations $\Gamma^{\star}$). *A structural representation $\Gamma^{\star}(\vec{S}, \mathbf{A}, \boldsymbol{X})$ of a non-empty subset $S \in \mathcal{P}^{\star}(V)$, of a graph $(\mathbf{A}, \boldsymbol{X}) \in \Sigma_n$, is most-expressive iff, there exists a bijective measurable map between $\Gamma^{\star}(\vec{U}, \mathbf{A}, \boldsymbol{X})$ and the orbit of $U$ in $G$ (Definition 7), $\forall U \in \mathcal{P}^{\star}(V)$ and $\forall (\mathbf{A}, \boldsymbol{X}) \in \Sigma_n$.*

Note, however, the failure to represent the link (*lynx*, *coyote*) in Figure 1 using the most-expressive node representations of the *lynx* and the *coyote*. A link needs to be represented by a joint representation of two nodes. For instance, we can easily verify from Definition 11 that $\Gamma^{\star}((\mathit{lynx,coyote}), \mathbf{A}, \boldsymbol{X}) \neq \Gamma^{\star}((\mathit{orca,coyote}), \mathbf{A}, \boldsymbol{X})$, even though $\Gamma^{\star}(\mathit{lynx}, \mathbf{A}, \boldsymbol{X}) = \Gamma^{\star}(\mathit{orca}, \mathbf{A}, \boldsymbol{X})$.

Next we show that joint prediction tasks only require joint structural representations. But first we need to show that any causal model defined through axiomatic counterfactuals (Galles & Pearl, 1998) can be equivalently defined through noise outsourcing (Austin, 2008), a straightforward result that we were unable to find in the literature.

**Lemma 2** (Causal modeling through noise outsourcing). *Definition 1 of Galles & Pearl (1998) gives a causal model as a triplet*

$$M = \langle U, V', F \rangle,$$

*where $U$ is a set of exogenous variables, $V'$ is a set of endogenous variables, and $F$ is a set of functions, such that in the causal model, $v_i' = f(\vec{pa}_i, u)$ is the realization of random variable $V_i' \in V'$ and a sequence of random variables $\vec{PA}_i$ with $PA_i \subseteq V \backslash V_i'$ as the endogenous variable parents of variable $V_i'$ as given by a directed acyclic graph. Then, there exists a pure random noise $\epsilon$ and a set of (measurable) functions $\{g_u\}_{u \in U}$ such that for $V_i \in V'$, $V_i'$ can be equivalently defined as $v_i' \overset{a.s.}{=} f(\vec{pa}_i, g_u(\epsilon_u))$, where $\epsilon_u$ has joint distribution $(\epsilon_u)_{\forall u \in U} \overset{a.s.}{=} g'(\epsilon)$ for some Borel measurable function $g'$ and a random variable $\epsilon \sim Uniform(0,1)$. The latter defines $M$ via noise outsourcing (Austin, 2008).*

The proof of Lemma 2 is given in the Appendix. Lemma 2 defines the causal model entirely via endogenous variables, deterministic functions, and a pure random noise random variable.

We are now ready for our theorem showing that joint prediction tasks only require (most-expressive) joint structural representations.

**Theorem 1.** *Let $\mathcal{S} \subseteq \mathcal{P}^{\star}(V)$ be a set of non-empty subsets of the vertices $V$. Let $\boldsymbol{Y}(\mathcal{S}, \mathbf{A}, \boldsymbol{X}) = (Y(\vec{S}, \mathbf{A}, \boldsymbol{X}))_{S \in \mathcal{S}}$ be a sequence of random variables defined over the sets $S \in \mathcal{S}$ of a graph $G = (\mathbf{A}, \boldsymbol{X})$, that are invariant to the ordering of $\vec{S}$, $S \in \mathcal{S}$, such that $Y(\vec{S}_1, \mathbf{A}, \boldsymbol{X}) \overset{d}{=} Y(\vec{S}_2, \mathbf{A}, \boldsymbol{X})$ for any two jointly isomorphic subsets $S_1, S_2 \in \mathcal{S}$ (Definition 7), where $\overset{d}{=}$ means equality in their marginal distributions. Then, there exists a measurable function $\varphi$ such that, $\boldsymbol{Y}(\mathcal{S}, \mathbf{A}, \boldsymbol{X}) \overset{a.s.}{=} (\varphi(\Gamma^{\star}(\vec{S}, \mathbf{A}, \boldsymbol{X}), \epsilon_S))_{S \in \mathcal{S}}$, where $\epsilon_S$ is the random noise that defines the exogenous variables of Lemma 2, with joint distribution $p((\epsilon_{S'})_{\forall S' \in \mathcal{S}})$ independent of $\mathbf{A}$ and $\boldsymbol{X}$.*

Theorem 1 extends Theorem 12 of Bloem-Reddy & Teh (2019) in multiple ways: (a) to all subsets of nodes, $S \in \mathcal{P}^{\star}(V)$, (b) to include causal language, and, most importantly, (c) to showing that any prediction task that can be defined over $S$, requires only a most-expressive joint structural representation over $S$. For instance, any task with $|S| = 2$ predicting a missing link $(u, v)$ on a graph $G = (\mathbf{A}, \boldsymbol{X})$, requires only the most-expressive structural representation $\Gamma^{\star}((u, v), \mathbf{A}, \boldsymbol{X})$. Note that, in order to predict directed edges, we must use $\Gamma^{\star}((u, v), \mathbf{A}, \boldsymbol{X})$ to also predict the edge's direction: $\to, \leftarrow, \leftrightarrow$, but a detailed procedure showing how to predict directed edges is relegated to a future journal version of this paper. Theorem 1 also includes node tasks for $|S| = 1$, hyperedge tasks for $2 < |S| < n$, and graph-wide tasks for $S = V$.

**Remark 1** (GNNs and link prediction). *Even though structural node representations of GNNs are not able to predict edges, GNNs are often still optimized to predict edges (e.g., (Hamilton et al., 2017a; Xu et al., 2018)) in transfer learning tasks. This optimization objective guarantees that any small topological differences between two nearly-isomorphic nodes without an edge will be amplified, while differences between nodes with an edge will be minimized. Hence, the topological differences in a close-knit community will be minimized in the representation. This procedure is an interesting way to introduce homophily in structural representations and should work well for node classification tasks in homophilic networks (where node classes tend to be clustered).*

We now turn our attention to node embeddings and their relationship with joint representations.

### 3.2 ON (POSITIONAL) NODE EMBEDDINGS

**Definition 12** (Node Embeddings). *The node embeddings of a graph $G = (\mathbf{A}, \boldsymbol{X})$ are defined as joint samples of random variables $(\boldsymbol{Z}_i)_{i \in V} | \mathbf{A}, \boldsymbol{X} \sim p(\cdot | \mathbf{A}, \boldsymbol{X})$, $\boldsymbol{Z}_i \in \mathbb{R}^d$, $d \geq 1$, where $p(\cdot | \mathbf{A}, \boldsymbol{X})$ is a $\mathcal{G}$-equivariant probability distribution on $\mathbf{A}$ and $\boldsymbol{X}$, that is, $\pi(p(\cdot | \mathbf{A}, \boldsymbol{X})) = p(\cdot | \pi(\mathbf{A}), \pi(\boldsymbol{X}))$ for any permutation $\pi \in \Pi_n$.*

Essentially, Definition 12 says that the probability distribution $p(\boldsymbol{Z} | \mathbf{A}, \boldsymbol{X})$ of a node embedding $\boldsymbol{Z}$ must be $\mathcal{G}$-equivariant on $\mathbf{A}$ and $\boldsymbol{X}$. This is the only property we require to define a node embedding. Next, we show that the node embeddings given by Definition 12 cover a wide range of embedding methods in the literature.

**Corollary 1.** *The node embeddings in Definition 12 encompass embeddings given by matrix and tensor factorization methods —such as Singular Value Decomposition (SVD), Non-negative Matrix Factorization (NMF), implicit matrix factorization (a.k.a. word2vec)–, latent embeddings given by Bayesian graph models —such as Probabilistic Matrix Factorizations (PMFs) and variants—, variational autoencoder methods and graph neural networks that use random lighthouses to extract node embedddings.*

The proof of Corollary 1 is in the Appendix, along with the references of each of the methods mentioned in the corollary. The output of some of the methods described in Corollary 1 is deterministic, and for those, the probability density $p(\boldsymbol{Z} | \mathbf{A}, \boldsymbol{X})$ is a Dirac delta. In practice, however, even deterministic methods use algorithms whose outputs depend on randomized initial conditions, which will also satisfy Corollary 1.

We now show that permutation equivariance implies two isomorphic nodes (or two subsets of nodes) must have the same marginal distributions over $\boldsymbol{Z}$:

**Lemma 3.** *The permutation equivariance of $p$ in Definition 12 implies that, if two proper subsets of nodes $S_1, S_2 \in \mathcal{P}^\star(V) \backslash V$ are isomorphic, then their marginal node embedding distributions must be the same up to a permutation, i.e., $p((\boldsymbol{Z}_i)_{i \in S_1} | \mathbf{A}, \boldsymbol{X}) = \pi(p((\boldsymbol{Z}_j)_{j \in S_2} | \mathbf{A}, \boldsymbol{X}))$ for some appropriate permutation $\pi \in \Pi_n$.*

Hence, the critical difference between the structural node representation vector $(\Gamma(v, \mathbf{A}, \boldsymbol{X}))_{v \in V}$ in Definition 8 and node embeddings $\boldsymbol{Z}$ in Definition 12, is that the vector $(\Gamma(v, \mathbf{A}, \boldsymbol{X}))_{v \in V}$ *must be $\mathcal{G}$-equivariant while $\boldsymbol{Z}$ need not be* —even though $\boldsymbol{Z}$'s distribution must be $\mathcal{G}$-equivariant. This seemly trivial difference has tremendous consequences, which we explore in the reminder of this section.

Next, we show that node embeddings $\boldsymbol{Z}$ cannot have any extra information about $G$ that is not already contained in a most-expressive structural representation $\Gamma^\star$.

**Theorem 2** (The statistical equivalence between node embeddings and structural representations). *Let $\boldsymbol{Y}(\mathcal{S}, \mathbf{A}, \boldsymbol{X}) = (Y(\vec{S}, \mathbf{A}, \boldsymbol{X}))_{S \in \mathcal{S}}$ be as in Theorem 1. Consider a graph $G = (\mathbf{A}, \boldsymbol{X}) \in \Sigma_n$, $n \geq 2$. Let $\Gamma^\star(\vec{S}, \mathbf{A}, \boldsymbol{X})$ be a most-expressive structural representation of nodes $S \in \mathcal{P}^\star(V)$ in $G$. Then,*

$$Y(\vec{S}, \mathbf{A}, \boldsymbol{X}) \perp\!\!\!\perp_{\Gamma^\star(\vec{S}, \mathbf{A}, \boldsymbol{X})} \boldsymbol{Z} | \mathbf{A}, \boldsymbol{X}, \quad \forall S \in \mathcal{S},$$

*for any node embedding matrix $\boldsymbol{Z}$ that satisfies Definition 12, where $A \perp\!\!\!\perp_B C$ means $A$ is independent of $C$ given $B$. Finally, $\forall (\mathbf{A}, \boldsymbol{X}) \in \Sigma_n$, there exists a most-expressive node embedding $\boldsymbol{Z}^\star | \mathbf{A}, \boldsymbol{X}$ such that,*

$$\Gamma^\star(\vec{S}, \mathbf{A}, \boldsymbol{X}) = \mathbb{E}_{\boldsymbol{Z}^\star}[f^{(|S|)}((\boldsymbol{Z}_v^\star)_{v \in S}) | \mathbf{A}, \boldsymbol{X}], \quad \forall S \in \mathcal{S},$$

*for some appropriate collection of functions* $\{f^{(k)}(\cdot)\}_{k=1,\dots,n}$.

The proof of Theorem 2 is given in the Appendix. Note that the most-expressive embedding $\boldsymbol{Z}^\star|\mathbf{A}, \boldsymbol{X}$ extends the insight used to make GNNs more expressive in Murphy et al. (2019) to a more general procedure.

Theorem 2 implies that, *for any graph prediction task, node embeddings carry no information beyond that of structural representations.* A less attentive reader may think this creates an apparent paradox, since one cannot predict a property $Y((\textit{lynx,coyote}), \mathbf{A}_{\text{food web}}, \boldsymbol{X}_{\text{food web}})$ in Figure 1 from structural node embeddings, since $\Gamma(\textit{lynx}, \mathbf{A}, \boldsymbol{X}) = \Gamma(\textit{orca}, \mathbf{A}, \boldsymbol{X})$. The resolution of the paradox is to note that Theorem 2 describes the prediction of a link through a pairwise structural representation $\Gamma((\textit{lynx,coyote}), \mathbf{A}_{\text{food web}}, \boldsymbol{X}_{\text{food web}})$, and we may not be able to do the same task with structural node representations alone. An interesting question for future work is how well can we learn distributions (representations) from (node embeddings) samples, extending (Kamath et al., 2015) to graph representations.

Other equally important consequences of Theorem 2 are: (a) any sampling approach obtaining node embeddings $\boldsymbol{Z}$ is valid as long as the distribution is $\mathcal{G}$-equivariant (Definition 12), noting that isomorphic nodes must have the same marginal distributions (per Lemma 3). (b) Interestingly, convex optimization methods for matrix factorization can be seen as variance-reduction techniques with no intrinsic value beyond reducing variance. (c) Methods that give unique node embeddings —if the embedding of any two isomorphic nodes are different— are provably incorrect when used to predict graph relationships since they are permutation-sensitive.

**Remark 2** (Some GNN methods give node embeddings not structural representations). *The random edges added by GraphSAGE (Hamilton et al., 2017a) and GIN (Xu et al., 2018) random walks make these methods node embeddings rather than structural node representations, according to Definition 12. To transform them back to structural node representations, one must average over all such random walks.*

The following corollaries describe other consequences of Theorem 2:

**Corollary 2.** *The link prediction task between any two nodes $u, v \in V$ depends only on the most-expressive tuple representation $\Gamma^\star((u, v), \mathbf{A}, \mathbf{X})$. Moreover, $\Gamma^\star((u, v), \mathbf{A}, \mathbf{X})$ always exists for any graph $(\mathbf{A}, \boldsymbol{X})$ and nodes $(u, v)$. Finally, given most-expressive node embeddings $\boldsymbol{Z}^\star$, there exists a function $f$ such that $\Gamma^\star((u, v), \mathbf{A}, \mathbf{X}) = \mathbb{E}_{\boldsymbol{Z}^\star}[f(\boldsymbol{Z}_u^\star, \boldsymbol{Z}_v^\star)], \forall u, v$.*

A generalization of Corollary 2 is also possible, where Theorem 2 is used to allow us to create joint representations from simpler node embedding sampling methods.

**Corollary 3.** *Sample $\boldsymbol{Z}$ according to Definition 12. Then, we can learn a $k$-node structural representation of a subset of $k$ nodes $S \in \mathcal{P}^\star(V)$, $|S| = k$, simply by learning a function $f^{(k)}$ whose average $\Gamma(\vec{S}, \mathbf{A}, \boldsymbol{X}) = \mathbb{E}[f^{(k)}((Z_v)_{v \in S})]$ can be used to predict $Y(\vec{S}, \mathbf{A}, \boldsymbol{X})$.*

The proof of Corollary 3 is in the Appendix. Finally, we show that the concepts of transductive and inductive learning are unrelated to the notions of node embeddings and structural representations.

**Corollary 4.** *Transductive and inductive learning are unrelated to the concepts of node embeddings and structural representations.*

Corollary 4 clears a confusion that, we believe, arises because traditional applications of node embeddings use a single Monte Carlo sample of $\boldsymbol{Z}|\mathbf{A}, \boldsymbol{X}$ to produce a structural representation (e.g., (Mikolov et al., 2013)). Inherently, a classifier learned with such a poor structural representation may fail to generalize over the test data, and will be deemed *transductive*.

**Corollary 5.** *Merging a node embeddings sampling scheme with GNNs can increase the structural representation power of GNNs.*

Corollary 5 is a direct consequence of Theorem 2, with Murphy et al. (2019) showing RP-GNN as a concrete method to do so.

## 4 RESULTS

This section focuses on applying the lessons learned in Section 3 in four tasks, divided into two common goals. The goal of the first three tasks is to show that, as described in Theorem 2, node embeddings can be used to create expressive structural embeddings of nodes, tuples, and triads. These

representations are then subsequently used to make predictions on downstream tasks with varied node set sizes. The tasks also showcase the added value of using multiple node embeddings (Monte Carlo) samples to estimate structural representations, both during training and testing. Moreover, showcasing Theorem 1 and the inability of node representations to capture joint structural representations, these tasks show that structural node representations are useless in prediction tasks over more than one node, such as links and triads. The goal of fourth task is to showcase how multiple Monte Carlo samples of node embeddings are required to observe the fundamental relationship between structural representations and node embeddings predicted by Theorem 2.

*An important note:* Our proposed theoretical framework is not limited to the way we generate node embeddings. For example, our theoretical framework can use SVD in an inductive setting, where we train a classifier in one graph and test in a different graph, which was thought not possible previously with SVD. SVD with our theoretical framework is denoted MC-SVD, to emphasize the importance of Monte Carlo sampling in building better structural representations. Alternatively, more expressive node embeddings can be obtained using Colliding Graph Neural Networks (CGNN), as we show in the Appendix (Section 10 and Section 11)

## 4.1 QUANTITATIVE RESULTS

In what follows, we evaluate structural representations estimated from four node embedding techniques, namely GIN (Xu et al., 2018), RP-GIN (Murphy et al., 2019), 1-2-3 GNN (Morris et al., 2019), MC-SVD and CGNN. We classify GIN, RP-GIN and 1-2-3 GNN as node embedding techniques, as they employ the unsupervised learning procedure of Hamilton et al. (2017a). These were chosen because of their potential extra link and triad representation power over traditional structural representation GNNs. All node embedding methods are evaluated by their effect in estimating good structural representation for downstream task accuracy. We partition $G = (\mathbf{A}, \boldsymbol{X})$ into three non-overlapping induced subgraphs, namely $G_{\text{train}} = (\mathbf{A}_{\text{train}}, \boldsymbol{X}_{\text{train}})$, $G_{\text{val}} = (\mathbf{A}_{\text{val}}, \boldsymbol{X}_{\text{val}})$ and $G_{\text{test}} = (\mathbf{A}_{\text{test}}, \boldsymbol{X}_{\text{test}})$, which we use for training, validation and testing, respectively. In learning all four node embedding techniques, we only make use of the graphs $G_{\text{train}}$ and $G_{\text{val}}$. All the four models used here have never seen the test graph $G_{\text{test}}$ before test time —i.e., all our node embedding methods, used in the framework of Theorem 2, behave like inductive methods.

*Monte Carlo joint representations during an* **unsupervised** *learning phase:* A key component of our optimization is learning joint representations from node embeddings —as per Theorem 2. For this, at each gradient step (in practice, we do at each epoch), we perform a Monte Carlo sample of the node embeddings $\boldsymbol{Z}|\mathbf{A}, \boldsymbol{X}$. This, procedure optimizes a proper upper bound on the empirical loss, if the loss is the negative log-likelihood, cross-entropy, or a square loss. The proof is trivial by Jensen's inequality. For GIN, RP-GIN and 1-2-3 GNN, we add random edges to the graph following a random walk at each epoch (Hamilton et al., 2017a). For the MC-SVD procedure, we use the left eigenvector matrix obtained by: (1) a random seed, (2) a random input permutation of the adjacency matrix, and (3) a single optimization step, rather than running SVD until it converges. We also have results with MC-SVD[†], which is the same procedure as before, but runs SVD until convergence —noting that the latter is likely to give deterministic results in large real-world graphs.

*Monte Carlo joint representations during a* **supervised** *learning phase:* During the supervised phase, we first estimate a structural joint representation $\hat{\Gamma}(\vec{S}, \mathbf{A}, \boldsymbol{X})$ as the average of $m \in \{1, 5, 20\}$ Monte Carlo samples of a permutation-invariant function (Murphy et al., 2018; Zaheer et al., 2017) (sum-pooling followed by an MLP) applied to a sampled node embedding $(\boldsymbol{Z}_v)_{v \in S}|\mathbf{A}, \boldsymbol{X}$. Then, using $\hat{\Gamma}(\vec{S}, \mathbf{A}, \boldsymbol{X})$, we predict the corresponding target variable $Y(S, \mathbf{A}, \boldsymbol{X})$ of each task using an MLP. The node sets of our tasks $S \subseteq V$, have sizes $|S| \in \{1, 2, 3\}$, corresponding to node classification, link prediction, and triad prediction tasks, respectively.

*Datasets:* We consider four graph datasets used by Hamilton et al. (2017a), namely Cora, Citeseer, Pubmed (Namata et al., 2012; Sen et al., 2008) and PPI (Zitnik & Leskovec, 2017). Cora, Citeseer and Pubmed are citation networks, where vertices represent papers, edges represent citations, and vertex features are bag-of-words representation of the document text. The PPI (protein-protein interaction) dataset is a collection of multiple graphs representing the human tissue, where vertices represent proteins, edges represent interactions across them, and node features include genetic and immunological information. Train, validation and test splits are used as proposed by Yang et al. (2016) (see Table 3 in the Appendix). Further dataset details can be found in the Appendix.

Table 1: Micro F1 score on three distinct tasks averaged over 12 runs with standard deviation in parenthesis. The number within the parenthesis beside the model name indicates the number of Monte Carlo samples used in the estimation of the structural representation. MC-SVD$^\dagger$(1) denotes the SVD procedure run until convergence with one Monte Carlo sample for the representation. Bold values show maximum empirical average, and multiple bolds happen when its standard deviation overlaps with another average. Results for Citeseer are provided in the Appendix in Table 2.

| | Node Classification | | | Link Prediction | | | Triad Prediction | | |
|---|---|---|---|---|---|---|---|---|---|
| | Cora | Pubmed | PPI | Cora | Pubmed | PPI | Cora | Pubmed | PPI |
| *Random* | 0.143 | 0.333 | $0.5^{121}$ | 0.500 | 0.500 | 0.500 | 0.250 | 0.250 | 0.250 |
| GIN(1) | 0.646(0.021) | **0.878(0.006)** | 0.533(0.003) | 0.526(0.029) | 0.513(0.048) | 0.604(0.018) | 0.280(0.010) | 0.430(0.019) | 0.400(0.006) |
| GIN(5) | 0.676(0.031) | **0.880(0.003)** | 0.535(0.004) | 0.491(0.019) | 0.517(0.028) | 0.609(0.012) | 0.284(0.017) | 0.422(0.024) | 0.397(0.004) |
| GIN(20) | 0.678(0.024) | **0.880(0.002)** | 0.536(0.003) | 0.514(0.026) | 0.512(0.042) | 0.603(0.010) | 0.281(0.010) | 0.422(0.028) | 0.399(0.004) |
| RP-GIN(1) | 0.655(0.023) | **0.879(0.002)** | 0.534(0.005) | 0.506(0.016) | 0.616(0.048) | 0.605(0.011) | 0.283(0.013) | 0.423(0.024) | 0.400(0.005) |
| RP-GIN(5) | 0.681(0.022) | **0.881(0.004)** | 0.534(0.004) | 0.498(0.016) | 0.637(0.038) | 0.612(0.006) | 0.285(0.025) | 0.429(0.024) | 0.399(0.009) |
| RP-GIN(20) | 0.675(0.032) | **0.879(0.005)** | 0.533(0.003) | 0.518(0.017) | 0.619(0.032) | 0.603(0.007) | 0.279(0.011) | 0.418(0.011) | 0.393(0.003) |
| 1-2-3 GNN(1) | 0.319(0.017) | 0.412(0.005) | 0.403(0.003) | 0.501(0.007) | 0.495(0.018) | 0.502(0.005) | 0.280(0.010) | 0.416(0.020) | 0.250(0.003) |
| 1-2-3 GNN(5) | 0.321(0.008) | 0.395(0.065) | 0.405(0.001) | 0.501(0.018) | 0.500(0.002) | 0.501(0.003) | 0.285(0.015) | 0.418(0.029) | 0.251(0.005) |
| 1-2-3 GNN(20) | 0.324(0.010) | 0.462(0.113) | 0.401(0.007) | 0.501(0.007) | 0.499(0.002) | 0.501(0.008) | 0.285(0.014) | 0.419(0.026) | 0.254(0.008) |
| MC-SVD$^\dagger$(1) | 0.665(0.014) | 0.810(0.009) | 0.523(0.005) | 0.588(0.029) | 0.807(0.024) | **0.755(0.010)** | 0.336(0.038) | 0.515(0.077) | 0.532(0.010) |
| MC-SVD(1) | 0.667(0.017) | 0.825(0.007) | 0.521(0.006) | 0.583(0.020) | 0.818(0.032) | **0.755(0.008)** | 0.304(0.034) | 0.518(0.065) | 0.529(0.006) |
| MC-SVD(5) | 0.669(0.013) | 0.842(0.015) | 0.556(0.009) | 0.572(0.019) | **0.848(0.038)** | **0.754(0.006)** | 0.306(0.037) | **0.567(0.061)** | **0.544(0.008)** |
| MC-SVD(20) | 0.672(0.013) | 0.855(0.010) | 0.591(0.009) | 0.580(0.021) | **0.868(0.029)** | **0.762(0.010)** | 0.300(0.033) | **0.546(0.029)** | **0.550(0.007)** |
| CGNN(1) | 0.468(0.026) | 0.686(0.020) | 0.545(0.010) | 0.682(0.026) | 0.587(0.027) | 0.661(0.015) | 0.352(0.028) | 0.404(0.014) | 0.414(0.009) |
| CGNN(5) | 0.641(0.022) | 0.808(0.008) | 0.637(0.014) | **0.707(0.027)** | 0.585(0.037) | 0.704(0.012) | **0.414(0.045)** | 0.417(0.018) | 0.463(0.026) |
| CGNN(20) | **0.726(0.024)** | 0.831(0.010) | **0.707(0.015)** | **0.712(0.041)** | 0.581(0.039) | 0.738(0.011) | **0.405(0.034)** | 0.419(0.017) | 0.498(0.021) |

*Node classification task:* This task predicts node classes for each of the four datasets. In this task, structural node representations are enough. The structural node representation is used to classify nodes into different classes using an MLP, whose weights are trained in a supervised manner using the same splits as described above. In Cora, Citeseer and Pubmed, each vertex belongs only to a single class, whereas in the PPI graph dataset, nodes could belong to multiple classes.

*Link prediction task:* Here, we predict a small fraction of edges and non-edges in the test graph, as well as identify all false edges and non-edges (which were introduced as a corruption of the original graph) between different pairs of nodes in the graph. Specifically, we use joint tuple representations $\Gamma((u,v), \mathbf{A}, \boldsymbol{X})$, for $u, v \in V$, as prescribed by Theorem 2. Since, datasets are sparse in nature, and a trivial 'non-edges' predictor would result in a very high accuracy, we balance the train and validation and test splits to contain an equal number of edges and non-edges.

*Triad prediction task:* This task involves the prediction of triadic interaction as well as identification of possible fake interactions in the data between the three nodes under consideration. In this case, we use joint triadic representations $\Gamma((u,v,h), \mathbf{A}, \boldsymbol{X})$, for $u, v, h \in V$, as prescribed by Theorem 2. Here, we ensure that edge corruptions are dependent. We treat the graphs as being undirected in accordance with previous literature, and predict the number of true (uncorrupted) edges between the three nodes. Again, to handle the sparse nature of the graphs, we use a balanced dataset for train, validation, and test.

In Table 1 we present Micro-F1 scores for all four models over the three tasks. First, we note how more Monte Carlo samples at test time tend to increase test accuracy. In *node classification tasks*, we note that structural node representations from CGNN node embeddings significantly outperform other methods in two of the three datasets (the harder tasks). In *link prediction tasks*, the low accuracy of GNN-based methods (close to random) showcases the little extra-power of GIN and RP-GIN sampling schemes have over the the inability of structural node representations to predict links. Surprisingly, in *triads predictions*, the accuracy of GNN-based methods is much above random in some datasets, but still far from other node embedding methods. In *link and triad prediction tasks*, MC-SVD and CGNN share the lead with MC-SVD winning on Pubmed and PPI, and CGNN being significantly more accurate on Cora. Although, the 1-2-3 GNN is based on the two-Weisfeiler-Lehman (2-WL) algorithm (pairwise Weisfeiler-Lehman algorithm (Fürer, 2017)), which provides tuple representations that can be exploited towards link prediction, it is however an approximation primarily designed for graph classification tasks. Unfortunately, the 1-2-3 GNN performs quite poorly on all our tasks (node classification, link and triad prediction), indicating the need for a task-specific approximation of 2-WL GNN's. Results for Citeseer, in the Appendix, show similar results.

## 4.2 QUALITATIVE RESULTS

We now investigate the transformation of node embedding into node and link structural representations. Theorem 2 shows that the average of a function over node embedding Monte Carlo samples gives node and link embedding. In this experiment, we empirically test Theorem 2, by creating structural representations from the node embedding random matrix $\boldsymbol{Z}$, defined as the left eigenvector matrix obtained through SVD (ran until convergence), with the sources of randomness being due to a random permutation of the adjacency matrix given as input to the SVD method and the random seed it uses. Consider $m$ such embedding matrices Monte Carlo samples, $\mathcal{Z}^{(m)} = \{\boldsymbol{Z}^{(i)}\}_{i=1}^{m}$.

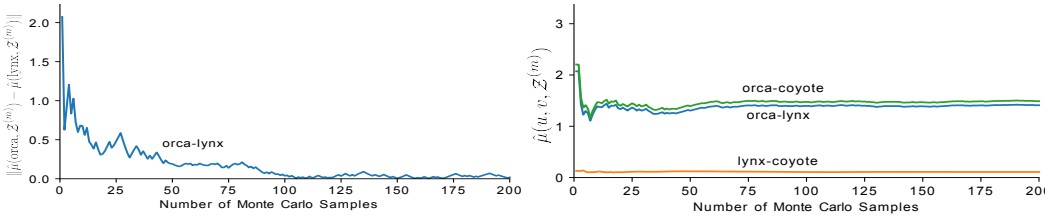

(a) Difference in avg. structural node representations.   (b) Average structural link representations

Figure 2: Structural Representations for nodes and links using multiple samples obtained using MC-SVD on the disconnected food web graph shown in Figure 1.

*Structural node representations from node embeddings:* According to Theorem 2, the average $\mathbb{E}[\boldsymbol{Z}_{v,\cdot}|\mathbf{A}]$ is a valid structural representation of node $v \in V$ in the adjacency matrix $\mathbf{A}$ of Figure 1. To test this empirically, we consider the unbiased estimator $\hat{\mu}(v, \mathcal{Z}^{(m)}) = \frac{1}{m}\sum_{i=1}^{m} \boldsymbol{Z}_{v,\cdot}^{(i)}$, $v \in V$, where $\lim_{m\to\infty} \hat{\mu}(v, \mathcal{Z}^{(m)}) \stackrel{a.s.}{=} \mathbb{E}[\boldsymbol{Z}_{v,\cdot}|\mathbf{A}]$. Figure 2a shows the Euclidean distance between the empirical structural representations $\hat{\mu}(\text{orca}, \mathcal{Z}^{(m)})$ and $\hat{\mu}(\text{lynx}, \mathcal{Z}^{(m)})$ as a function of $m \in [1, 200]$. As expected, because these two nodes are isomorphic, $\|\hat{\mu}(\text{orca}, \mathcal{Z}^{(m)}) - \hat{\mu}(\text{lynx}, \mathcal{Z}^{(m)})\| \to 0$ as $m$ grows, with $m = 100$ giving reasonably accurate results.

*Structural link representations from node embeddings:* According to Theorem 2, the average $\mathbb{E}[f^{(2)}(\boldsymbol{Z}_{u,\cdot}, \boldsymbol{Z}_{v,\cdot})|\mathbf{A}]$ of a function $f^{(2)}$ is a valid structural representation of a link with nodes $u, v \in V$ in the adjacency matrix $\mathbf{A}$ of Figure 1. As an example, we use $f^{(2)}(a, b) = \|a - b\|$, and define the unbiased estimator $\hat{\mu}(u, v, \mathcal{Z}^{(m)}) = \frac{1}{m}\sum_{i=1}^{m}\|Z_u^{(i)} - Z_v^{(i)}\|, \forall u, v \in V$, where $\lim_{m\to\infty} \hat{\mu}(u, v, \mathcal{Z}^{(m)}) \stackrel{a.s.}{=} \mathbb{E}[\|\boldsymbol{Z}_{u,\cdot} - \boldsymbol{Z}_{v,\cdot}\||\mathbf{A}]$. Figure 2b shows the impact of increasing the number of Monte Carlo samples $m$ over the empirical structural representation of links. We observe that although the empirical node representations of the orca and the lynx seem to converge to the same value, $\lim_{m\to\infty} \hat{\mu}(\text{orca}, \mathcal{Z}^{(m)}) = \lim_{m\to\infty} \hat{\mu}(\text{lynx}, \mathcal{Z}^{(m)})$, their empirical joint representations with the coyote converge to different values, $\lim_{m\to\infty} \hat{\mu}(\text{lynx}, \text{coyote}, \mathcal{Z}^{(m)}) \neq \lim_{m\to\infty} \hat{\mu}(\text{orca}, \text{coyote}, \mathcal{Z}^{(m)})$, as predicted by Theorem 2. Also note a similar (but weaker) trend for $\lim_{m\to\infty} \hat{\mu}(\text{orca}, \text{lynx}, \mathcal{Z}^{(m)}) \neq \lim_{m\to\infty} \hat{\mu}(\text{orca}, \text{coyote}, \mathcal{Z}^{(m)})$, showing these three tuples to be structurally different.

## 5 RELATED WORK

*Node Embeddings vs Structural Representations:* Prior works have categorized themselves as one of either node embedding methods or methods which learn structural representations. This artificial separation, consequently led to little contemplation of the relation between the two, restricting each of these approaches to a certain subsets of downstream tasks on graphs. Node embeddings were arguably first defined in 1904, through Spearman's common factors. Ever since, there has never been a universal definition of node embedding: node embeddings were simply the product of a particular method. This literature features a myriad of methods, e.g., matrix factorization (Belkin & Niyogi, 2002; Cao et al., 2015; Ahmed et al., 2013; Ou et al., 2016), implicit matrix factorization (Mikolov et al., 2013; Arora et al., 2009; Chen et al., 2012; Perozzi et al., 2014; Grover & Leskovec, 2016), Bayesian factor models (Mnih & Salakhutdinov, 2008; Gopalan et al., 2014a), and some types of neural networks (You et al., 2019; Liang et al., 2018; Tang et al., 2019; Grover et al., 2019).

Arguably, the most common interpretation of node embeddings borrows from definitions of graph (node) embeddings in metric spaces: a measure of relative node *closeness* (Abraham et al., 2006; Bourgain, 1985; Candès & Recht, 2009; Graham & Winkler, 1985; Kleinberg, 2007; Linial et al., 1995; Rabinovich & Raz, 1998; Recht et al., 2010; Shaw et al., 2011). Even in non-metric methods, such as word2vec (Mikolov et al., 2013) and Glove (Pennington et al., 2014), the embeddings have properties similar to those of metric spaces (Nematzadeh et al., 2017). Note that the definition of *close* varies from method to method, i.e., it is *model-dependent*. Still, this interpretation of *closeness* is the reason why their downstream tasks are often link prediction and clustering. However, once the literature started defining relative node closeness with respect to structural neighborhood similarities (e.g., Henderson et al. (2012); Ribeiro et al. (2017); Donnat et al. (2018)), node embeddings and structural representations became more strangely entangled.

Structural representations have an increasing body of literature focused on node and whole-graph classification tasks. Theoretically, these works abandon metric spaces in favor of a group-theoretic

description of graphs (Bloem-Reddy & Teh, 2019; Chen et al., 2019; Kondor & Trivedi, 2018; Maron et al., 2019; Murphy et al., 2019), with connections to finite exchangeability and prior work on multilayer perceptrons (Wood & Shawe-Taylor, 1996). Graph neural networks (GNNs) (e.g., (Duvenaud et al., 2015; Gilmer et al., 2017; Kipf & Welling, 2016a; Hamilton et al., 2017a; Xu et al., 2018; Scarselli et al., 2008) among others) exploit this approach in tasks such as node and whole-graph classification. Morris et al. (2019) proposes a higher-order Weisfeller-Lehman GNN (WL-$k$-GNN), which is shown to get better accuracy in graph classification tasks than traditional (WL-1) GNNs. Unfortunately, Morris et al. (2019) focused only on graph-wide tasks, missing the fact that WL-2 GNN should be able to also perform link prediction tasks (Theorem 1), unlike WL-1 GNNs. More recently, graph neural networks have also been employed towards relational reasoning as well as matrix completion tasks (Schlichtkrull et al., 2018; Zhang & Chen, 2018; Battaglia et al., 2018; Berg et al., 2017; Monti et al., 2017). However, these GNN's, in general, learn node embeddings rather than structural node representations, which are then exploited towards link prediction. GNN-like architectures have been used to simulate dynamic programming algorithms (Xu et al., 2019), which is unrelated to graphs and outside the scope of this work.

To the best of our knowledge, our work is the first to provide the theoretical foundations connecting node embeddings and structural representations. A few recent works have classified node embedding and graph representation methods arguing them to be fundamentally different (e.g., Hamilton et al. (2017b); Rossi et al. (2019); Wu et al. (2019); Zhou et al. (2018)). Rather, our work shows that these are actually equivalent for downstream classification tasks, with the difference being that one is a Monte Carlo method (embedding) and the other one is deterministic (representation).

*Inductive vs Transductive Approaches:* Another common misconception our work uncovers, is that of qualifying node embedding methods as transductive learning and graph representation ones as inductive (e.g., Hamilton et al. (2017a); Yang et al. (2016)). In their original definitions, transductive learning (Gammerman et al., 1998), (Zhu et al., 2003), (Zhou et al., 2004) and inductive learning (Michalski, 1983), (Belkin et al., 2006) are only to be distinguished on the basis of generalizability of the learned model to unobserved instances. However, this has commonly been misinterpreted as node embeddings methods being transductive and structural representations being inductive. Models which depend solely only on the input feature vectors and the immediate neighborhood structure have been classified as inductive, whereas methods which rely on positional node embeddings to classify relationships in a graph have been incorrectly qualified as transductive.

The confusion seems to be rooted in researchers trying to use a single sample of a node embedding method and failing to generalize. Corollary 4 resolves this confusion by showing that transductive and inductive learning are fundamentally unrelated to positional node embeddings and graph representations. Both node embeddings and structural representations can be inductive if they can detect interesting conceptual patterns or reveal structure in the data. The theory provided by our work strongly adheres to this definition. Our work additionally provides the theoretical foundation behind the performance gains seen by Epasto & Perozzi (2019) and Goyal et al. (2019), which employ an ensemble of node embeddings for node classification tasks.

# 6 CONCLUSIONS

This work provided an invaluable unifying theoretical framework for node embeddings and structural graph representations, bridging methods like SVD and graph neural networks. Using invariant theory, we have shown (both theoretically and empirically) that relationship between structural representations and node embeddings is analogous to that of a distribution and its samples. We proved that all tasks that can be performed by node embeddings can also be performed by structural representations and vice-versa. Our empirical results show that node embeddings can be successfully used as inductive learning methods using our framework, and that non-GNN node embedding methods can be significantly more accurate in most tasks than simple GNNs methods. Our work introduced new practical guidelines to the use of node embeddings, which we expect will replace today's naïve direct use of node embeddings in graph tasks.

ACKNOWLEDGMENTS
This work was sponsored in part by the ARO, under the U.S. Army Research Laboratory contract number W911NF-09-2-0053, the Purdue Integrative Data Science Initiative and the Purdue Research foundation, the DOD through SERC under contract number HQ0034-13-D-0004 RT # 206, and the National Science Foundation under contract number CCF-1918483.

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

# APPENDIX

## 7 A VISUAL INTERPRETATION OF NODE EMBEDDINGS AND STRUCTURAL REPRESENTATIONS

In what follows we showcase the difference between structural representations (Figure 3) and positional node embeddings (Figure 4) obtained by graph neural networks and SVD, respectively, on the same graph. The graph is the food web of Figure 1 and consists of two disconnected subgraphs.

Figure 3 shows a 2-dimensional ($\mathbb{R}^2$) *structural representation* of nodes obtained by a standard GNN (obtained by the 1-WL GNN of Xu et al. (2018)), optimized to try to predict links without the addition of any random edges). Node colors represent the mapping of the learned structural representation in $\mathbb{R}^2$ into the red and blue interval of RGB space $[0, 255]^2$. Note that isomorphic nodes are forcibly mapped into the same structural representations (have the same color), even though the representation is trained to predict links. Specifically, the lynx and the orca in Figure 3 get the same color since they are isomorphic, while the lynx and the coyote have different structural representations (different colors). The lynx, like the orca, is a top predator in the food web while the coyote is not a top predator. Hence, it is quite evident why GNN's are not traditionally used for link prediction: as the structural node representation is a color, using these representations is akin to asking "is a light pink node in Figure 3 connected to a light blue node?" We cannot answer this question unless we also specify which light pink (i.e. coyote or seal) and which light blue (i.e. orca or lynx) we are talking about. Hence, the failure to predict links. Our Corollary 2 shows that joint 2-node structural representations are required for link prediction. Our Theorem 1 generalizes it to joint $k$-node structural representations for any $k$-node joint prediction task.

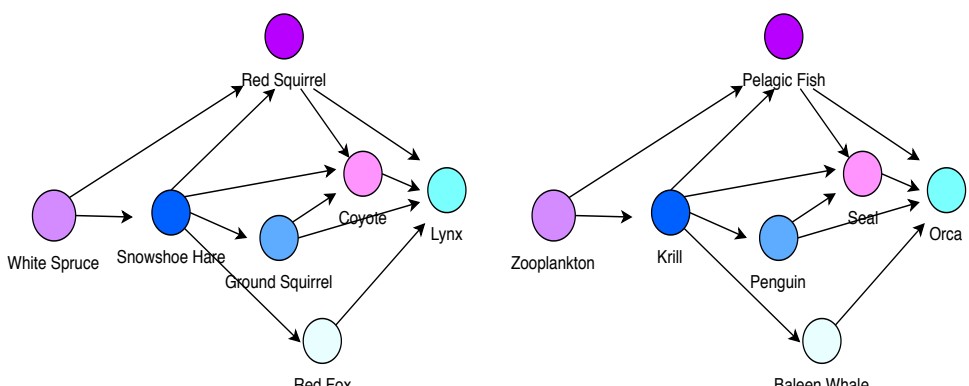

Figure 3: (Best in color) Food web graph of Figure 1 with node colors that map a two dimensional representation from a 1-WL GNN (GIN - (Xu et al., 2018)) into the $[0, 255]$ interval of blue and red intensity of RGB colors respectively. GIN is used a representative of structural representation of nodes. Structurally isomorphic nodes, obtain the same representation and are hence end up being visualised with the same color. Consequently, it is clear why structural node representations are used for node and graph classification, but not for link prediction.

Positional node embeddings, on the other hand, are often seen as a lower-dimensional projection of the rows and columns of the adjacency matrix $\boldsymbol{A}$ from $\mathbb{R}^n$ to $\mathbb{R}^d$, $d < n$, that preserves relative positions of the nodes in a graph. In Figure 4 we show the same food web graph, where now node colors map the values of the two leading (SVD) eigenvectors (of the undirected food web graph) into the $[0, 255]$ interval of blue and red intensity of RGB colors, respectively. The graph is made undirected, otherwise the left and right eigenvectors will be different and harder to represent visually. SVD is used here as a representative of positional node embedding methods. Note the lynx and the coyote now have close positional node embeddings (represented by colors which are closer in the color spectrum), while the positional node embeddings of the lynx and the orca are significantly different. Node embeddings are often seen as encoding the fact that the lynx and the coyote are part of a tightly-knit community, while the lynx and orca belong to distinct communities.

It is evident from Figure 4 why positional node embeddings are not traditionally used to predict node classes: predicting that the lynx, like the orca, is a top predator based on their node colors is

difficult, since the coyote's color is very similar to that of the lynx, while the orca obtains a completely different color to the lynx. Relying on color shades (light and dark) for node classification is unreliable, since nodes with completely different structural positions in the food chain may have similar color shades. Our Theorem 2 shows, however, that any positional node embedding (actually, any node embedding) must be a sample of a distribution given by the most-expressive structural representation of the node. That is, if SVD is ran again over different permutations of the vertices in the adjacency matrix (an isomorphic graph to Figure 1), the set of colors obtained by the lynx and the orca must be the same. Hence, node classification is possible if we look at the distribution of colors that a node obtains from random permutations of the adjacency matrix.

On the other hand, link prediction using the colors in Figure 4 is rather trivial, since similar colors mean "closeness" in the graph. For instance, we may easily predict that the baleen whale also eats zooplankton.

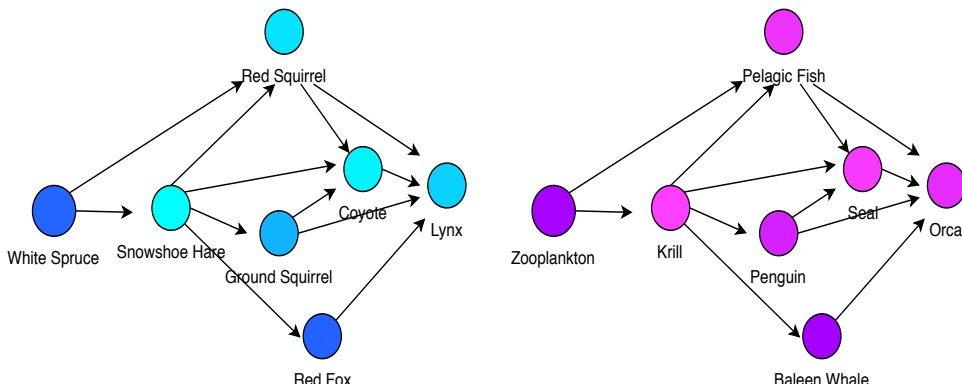

Figure 4: (Best in color) Food web graph of Figure 1 with node colors that map the values of the two leading (SVD) eigenvectors over the undirected graph into the $[0, 255]$ interval of blue and red intensity of RGB colors, respectively. SVD (run until convergence) is used as a representative of positional node embedding methods. The graph is made undirected, otherwise the left and right eigenvectors will be different and harder to represent visually. Note that nodes which are a part of the same connected component, obtain embeddings which are close in latent space, visually shown as similar colors. Consequently, it is clear that positional node embeddings can be used for link prediction and clustering.

## 8 PRELIMINARIES

**Noise outsourcing, representation learning, and graph representations:** The description of our proofs starts with the equivalence between probability models of graphs and graph representations. We start with the concept of noise outsourcing (Austin, 2008, Lemma 3.1) applied to our task —a weaker version of the more general concept of transfer (Kallenberg, 2006, Theorem 6.10) in pushforward measures.

A probability law $\boldsymbol{Z}|(\mathbf{A}, \boldsymbol{X}) \sim p(\cdot|(\mathbf{A}, \boldsymbol{X}))$, $(\mathbf{A}, \boldsymbol{X}) \in \Sigma_n$, can be described (Kallenberg, 2006, Theorem 6.10) by pure random noise $\epsilon \sim \text{Uniform}(0, 1)$ independent of $(\mathbf{A}, \boldsymbol{X})$, passing through a deterministic function $\boldsymbol{Z} = f((\mathbf{A}, \boldsymbol{X}), \epsilon)$ —where $f : \Sigma_n \times [0, 1] \to \Omega$, where $\Omega$ in our task will be a matrix $\Omega = \mathbb{R}^{n \times d}$ defining node representations, $d \geq 1$. That is, the randomness in the conditional $p(z|(\mathbf{A}, \boldsymbol{X}))$ is entirely outsourced to $\epsilon$, as $f$ is deterministic.

Now, consider replacing the graph $G = (\mathbf{A}, \boldsymbol{X})$ by a $\mathcal{G}$-equivariant representation $\Gamma(\mathbf{A}, \boldsymbol{X})$ of its nodes, the output of a neural network $\Gamma : \Sigma_n \to \mathbb{R}^{n \times m}$, $m \geq 1$, that gives an representation to each node in $G$. If a representation $\Gamma^\star(\mathbf{A}, \boldsymbol{X})$ is such that $\exists f'$ where $\boldsymbol{Z} = f(\mathbf{A}, \boldsymbol{X}, \epsilon) = f'(\Gamma^\star(\mathbf{A}, \boldsymbol{X}), \epsilon)$, $\forall (\mathbf{A}, \boldsymbol{X}) \in \Sigma_n$ and $\forall \epsilon \in [0, 1]$, then $\Gamma^\star(\mathbf{A}, \boldsymbol{X})$ does not lose any information when it comes to predicting $\boldsymbol{Z}$. Statistically (Kallenberg, 2006, Proposition 6.13), $\boldsymbol{Z} \perp\!\!\!\perp_{\Gamma^\star(\mathbf{A}, \boldsymbol{X})} (\mathbf{A}, \boldsymbol{X})$. We call $\Gamma^\star(\mathbf{A}, \boldsymbol{X})$ a most-expressive representation of $G$ with respect to $\boldsymbol{Z}$. A most-expressive representation (without qualifications) is one that is most-expressive for any target variable.

Representation learning is powerful precisely because it can learn functions $\Gamma$ that are compact and can encode most of the information available on the input. And because the most-expressive $\Gamma^\star$ is $\mathcal{G}$-equivariant, it also guarantees that any $\mathcal{G}$-equivariant function over $\Gamma^\star$ that outputs $\boldsymbol{Z}$ is also $\mathcal{G}$-equivariant, without loss of information.

## 9 Proof of Theorems, Lemmas and Corollaries

We restate and prove **Lemma 1**

**Lemma 1.** *Two nodes $v, u \in V$, have the same most-expressive structural representation $\Gamma^\star(v, \mathbf{A}, \boldsymbol{X}) = \Gamma^\star(u, \mathbf{A}, \boldsymbol{X})$ iff $u$ and $v$ are isomorphic nodes in $G = (\mathbf{A}, \boldsymbol{X})$ (Definition 6).*

*Proof.* In this proof, we consider both directions.

($\Rightarrow$) Consider two nodes $v, u \in V$ which satisfy the condition, $\Gamma^\star(v, \mathbf{A}, \boldsymbol{X}) = \Gamma^\star(u, \mathbf{A}, \boldsymbol{X})$ but are not isomorphic in $G = (\mathbf{A}, \boldsymbol{X})$. By contradiction, suppose $u$ and $v$ have different node orbits. This is a contradiction, since the bijective mapping of Definition 9 would have to take the same input and map them to different outputs.

($\Leftarrow$) By contradiction, consider the two nodes $u, v \in V$ which are isomorphic in $G = (\mathbf{A}, \boldsymbol{X})$ but with different most expressive structural representations i.e. $\Gamma^\star(v, \mathbf{A}, \boldsymbol{X}) \neq \Gamma^\star(u, \mathbf{A}, \boldsymbol{X})$. This is a contradiction, because as per Definition 6 two nodes should have the same structural representation, which would imply the most expressive structural representation is not a structural representation. Hence, the two nodes should share the same most expressive structural representation. $\square$

Next, we restate and prove Lemma 2

**Lemma 2** (Causal modeling through noise outsourcing). *Definition 1 of Galles & Pearl (1998) gives a causal model as a triplet*
$$M = \langle U, V, F \rangle,$$
*where $U$ is a set of exogenous variables, $V$ is a set of endogenous variables, and $F$ is a set of functions, such that in the causal model, $v_i = f(\vec{pa}_i, u)$ is the realization of random variable $V_i \in V$ and a sequence of random variables $\vec{PA}_i$ with $PA_i \subseteq V \backslash V_i$ as the endogenous variable parents of variable $V_i$ as given by a directed acyclic graph. Then, there exists a pure random noise $\epsilon$ and a set of (measurable) functions $\{g_u\}_{u \in U}$ such that for $V_i \in V$, $V_i$ can be equivalently defined as $v_i \overset{a.s.}{=} f(\vec{pa}_i, g_u(\epsilon_u))$, where $\epsilon_u$ has joint distribution $(\epsilon_u)_{\forall u \in U} \overset{a.s.}{=} g'(\epsilon)$ for some Borel measurable function $g'$ and a random variable $\epsilon \sim \text{Uniform}(0, 1)$. The latter defines $M$ via noise outsourcing (Austin, 2008).*

*Proof.* By Definition 1 of Galles & Pearl (1998), the set of exogenous variables $U$ are not affected by the set of endogeneous variables $V$. The noise outsourcing lemma (Lemma 3.1) of Austin (2008) (or its more complete version Theorem 6.10 of Kallenberg (2006)) shows that any samples of a joint distribution over a set of random variables $U$ can be described as $(u)_{\forall u \in U} \overset{a.s.}{=} g(\epsilon)$, for some Borel measurable function $g$ and a random variable $\epsilon \sim \text{Uniform}(0, 1)$. As the composition of two Borel measurable functions is also Borel measurable, it is trivial to show that there exists Borel measurable functions $\{g_u\}_{u \in U}$ and $g'$, such that $u \overset{a.s.}{=} g_u(\epsilon_u)$ and $(\epsilon_u)_{\forall u \in U} \overset{a.s.}{=} g'(\epsilon)$. The latter is trivial since $g_u$ can just be the identity function, $g_u(x) = x$. $\square$

Next, we restate and prove **Lemma 3**

**Lemma 3.** *The permutation equivariance of $p$ in Definition 12 implies that, if two proper subsets of nodes $S_1, S_2 \subsetneq V$ are isomorphic, then their marginal node embedding distributions must be the same up to a permutation, i.e., $p((\boldsymbol{Z}_i)_{i \in S_1} | \mathbf{A}, \boldsymbol{X}) = \pi(p((\boldsymbol{Z}_j)_{j \in S_2} | \mathbf{A}, \boldsymbol{X}))$ for some appropriate permutation $\pi$.*

*Proof.* From Definition 12, it is trivial to observe that two isomorphic nodes $u, v \in V$ in graph $G = (\mathbf{A}, \boldsymbol{X})$ have the same marginal node embedding distributions. In this proof we extend this to node sets $S \subset V$ where $|S| > 1$. We marginalize over $(Z_i)_{i \notin S_1}$ to obtain $p((\boldsymbol{Z}_i)_{i \in S_1} | \mathbf{A}, \boldsymbol{X})$ and in the other case over $(Z_i)_{i \notin S_2}$ to obtain $p((\boldsymbol{Z}_i)_{i \in S_2} | \mathbf{A}, \boldsymbol{X})$ respectively.

We consider 2 cases as follows:

Case 1: $S_1 = S_2$: This is the trivial where $S_1$ and $S_2$ are the exact same nodes, hence their marginal distributions are the identical as well by definition.

Case 2: $S_1 \neq S_2$: Since $S_1$ and $S_2$ are also given to be isomorphic, it is clear to see that every node in $S_1$ has an isomorphic equivalent in $S_2$. In a graph $G = (\mathbf{A}, \boldsymbol{X})$, the above statement conveys that $S_2$ can be written as a permutation $\pi$ on $S_1$, i.e $S_2 = \pi(S_1)$. Now, employing Definition 12, it is clear to see that $p((\boldsymbol{Z}_i)_{i \in S_1} | \mathbf{A}, \boldsymbol{X}) = \pi(p((\boldsymbol{Z}_j)_{j \in S_2} | \mathbf{A}, \boldsymbol{X}))$ ☐

Next, we restate and prove **Theorem 1**

**Theorem 1.** *Let $\mathcal{S} \subseteq \mathcal{P}(V)$ be a set of subsets of $V$. Let $\boldsymbol{Y}(\mathcal{S}, \mathbf{A}, \boldsymbol{X}) = (Y(\vec{S}, \mathbf{A}, \boldsymbol{X}))_{S \in \mathcal{S}}$ be a sequence of random variables defined over the sets $S \in \mathcal{S}$ of a graph $G = (\mathbf{A}, \boldsymbol{X})$, such that we define $Y(\vec{S}, \mathbf{A}, \boldsymbol{X}) := Y_{\vec{S}} | \mathbf{A}, \boldsymbol{X}$ and $Y(\vec{S}_1, \mathbf{A}, \boldsymbol{X}) \overset{d}{=} Y(\vec{S}_2, \mathbf{A}, \boldsymbol{X})$ for any two jointly isomorphic subsets $S_1, S_2 \in \mathcal{S}$ in $(\mathbf{A}, \boldsymbol{X})$ (Definition 7), where $\overset{d}{=}$ means equality in their marginal distributions. Then, there exists a deterministic function $\varphi$ such that, $\boldsymbol{Y}(\mathcal{S}, \mathbf{A}, \boldsymbol{X}) \overset{a.s.}{=} (\varphi(\Gamma^{\star}(\vec{S}, \mathbf{A}, \boldsymbol{X}), \epsilon_S))_{S \in \mathcal{S}}$, where $\epsilon_S$ is a pure source of random noise from a joint distribution $p((\epsilon_{S'})_{\forall S' \in \mathcal{S}})$ independent of $\mathbf{A}$ and $\boldsymbol{X}$.*

*Proof.* The case $S = V$ is given in Theorem 12 of Bloem-Reddy & Teh (2019). The case $S = \emptyset$ is trivial. The case $S \subsetneq V$, $S \neq \emptyset$, is described as follows with a constructive proof. First consider the case of two isomorphic sets of nodes $S_1, S_2 \in \mathcal{S}$. As by definition $Y(\vec{S}_1, \mathbf{A}, \boldsymbol{X}) \overset{d}{=} Y(\vec{S}_2, \mathbf{A}, \boldsymbol{X})$, we must assume $p(Y_{\vec{S}_1} | \mathbf{A}, \boldsymbol{X}) = p(Y_{\vec{S}_2} | \mathbf{A}, \boldsymbol{X})$. We can now use the transfer theorem (Kallenberg, 2006, Theorem 6.10) to obtain a joint description $Y_{\vec{S}_1} | \mathbf{A}, \boldsymbol{X} \overset{a.s.}{=} \phi_1(\vec{S}_1, \mathbf{A}, \boldsymbol{X}, \epsilon)$ and $Y_{\vec{S}_2} | \mathbf{A}, \boldsymbol{X} \overset{a.s.}{=} \phi_2(\vec{S}_2, \mathbf{A}, \boldsymbol{X}, \epsilon)$, where $\epsilon$ is a common source of independent noise. As $S_1$ and $S_2$ are joint isomorphic (Definition 7), there exists an isomorphism $S_1 = \text{iso}(\vec{S}_2)$, where $\phi_1(\text{iso}(\vec{S}_2), \mathbf{A}, \boldsymbol{X}, \epsilon) = \phi_1(\vec{S}_1, \mathbf{A}, \boldsymbol{X}, \epsilon)$. Because the distribution given by $\phi_1(\cdot, \epsilon)$ must be isomorphic-invariant in $(\mathbf{A}, \boldsymbol{X})$ and $S_1$ and $S_2$ are also isomorphic in $(\mathbf{A}, \boldsymbol{X})$ then, for all permutation actions $\pi \in \Pi_n$, there exists a new isomorphism $\text{iso}'$ such that $\phi_1(\pi(\text{iso}(\vec{S}_2)), \pi(\mathbf{A}), \pi(\boldsymbol{X}), \epsilon) \overset{d}{=} \phi_1(\text{iso}'(\pi(\vec{S}_1)), \pi(\mathbf{A}), \pi(\boldsymbol{X}), \epsilon)$, which allows us to create a function $\varphi'$ that incorporates $\text{iso}'$ into $\phi_1$. Due to the isomorphism between $S_1$ and $S_2$, we can do the same process for $S_2$ to arrive at the same function $\varphi'$. We can now apply Corollary 6.11 (Kallenberg, 2006) over $(Y_{\vec{S}_1} | \mathbf{A}, \boldsymbol{X}, Y_{\vec{S}_2} | \mathbf{A}, \boldsymbol{X})$ along with a measure-preserving mapping $f$ to show that $Y_{\vec{S}_1} | \mathbf{A}, \boldsymbol{X} \overset{a.s.}{=} \varphi'(\vec{S}_1, \mathbf{A}, \boldsymbol{X}, \epsilon_1)$ and $Y_{\vec{S}_2} | \mathbf{A}, \boldsymbol{X} \overset{a.s.}{=} \varphi'(\vec{S}_2, \mathbf{A}, \boldsymbol{X}, \epsilon_2)$, where $(\epsilon_1, \epsilon_2) = f(\epsilon)$. If $S_1$ and $S_2$ are not joint isomorphic, we can simply define $\varphi'(S_i, \cdot) := \phi_i(S_i, \cdot)$. Definition 11 allows us to define a function $\varphi$ from which we rewrite $\varphi'(\vec{S}_i, \mathbf{A}, \boldsymbol{X}, \epsilon_i)$ as $\varphi(\Gamma^{\star}(\vec{S}_i, \mathbf{A}, \boldsymbol{X}), \epsilon_i)$. Applying the same procedure to all $S \in \mathcal{S}$ concludes our proof. ☐

Next, we restate and prove **Theorem 2**

**Theorem 2** (The statistical equivalence between node embeddings and structural representations)**.** *Let $\boldsymbol{Y}(\mathcal{S}, \mathbf{A}, \boldsymbol{X}) = (Y(\vec{S}, \mathbf{A}, \boldsymbol{X}))_{S \in \mathcal{S}}$ be as in Theorem 1. Consider a graph $G = (\mathbf{A}, \boldsymbol{X}) \in \Sigma_n$. Let $\Gamma^{\star}(\vec{S}, \mathbf{A}, \boldsymbol{X})$ be a most-expressive structural representation of nodes $S \in \mathcal{S}$ in $(\mathbf{A}, \boldsymbol{X})$. Then,*

$$Y(\vec{S}, \mathbf{A}, \boldsymbol{X}) \perp\!\!\!\perp_{\Gamma^{\star}(\vec{S}, \mathbf{A}, \boldsymbol{X})} \boldsymbol{Z} | \mathbf{A}, \boldsymbol{X}, \quad \forall S \in \mathcal{S},$$

*for any node embedding matrix $\boldsymbol{Z}$ that satisfies Definition 12, where $A \perp\!\!\!\perp_B C$ means $A$ is independent of $C$ given $B$. Finally, $\forall (\mathbf{A}, \boldsymbol{X}) \in \Sigma_n$, there exists a most-expressive node embedding $\boldsymbol{Z}^{\star} | \mathbf{A}, \boldsymbol{X}$ such that,*

$$\Gamma^{\star}(\vec{S}, \mathbf{A}, \boldsymbol{X}) = \mathbb{E}_{\boldsymbol{Z}^{\star}}[f^{(|S|)}((\boldsymbol{Z}_v^{\star})_{v \in S}) | \mathbf{A}, \boldsymbol{X}], \quad \forall S \in \mathcal{S},$$

*for some appropriate collection of functions $\{f^{(k)}(\cdot)\}_{k=1,\dots,n}$.*

*Proof.* In the first part of the proof, for any embedding distribution $p(\boldsymbol{Z} | \mathbf{A}, \boldsymbol{X})$, we note that by Theorem 1, $y(\vec{S}, \mathbf{A}, \boldsymbol{X}) \overset{a.s.}{=} f'(\Gamma^{\star}(\vec{S}, \mathbf{A}, \boldsymbol{X}), \epsilon_S)$. Hence, $Y(\vec{S}, \mathbf{A}, \boldsymbol{X}) \perp\!\!\!\perp_{\Gamma^{\star}(\vec{S}, \mathbf{A}, \boldsymbol{X})} \boldsymbol{Z} | \mathbf{A}, \boldsymbol{X}, \forall S \in \mathcal{S}$, is a direct consequence of Proposition 6.13 in (Kallenberg, 2006).

In the second part of the proof, we construct an orbit over a most-expressive representation of a graph $(\mathbf{A}, \boldsymbol{X})$ of size $n$, with permutations that act only on unique node ids (node orderings) added as node features: $\Pi'(\mathbf{A}, \boldsymbol{X}) = \{((\Gamma^\star(v, \mathbf{A}, [\boldsymbol{X}, \pi(1, \ldots, n)^T]))_{\forall v \in V}\}_{\forall \pi \in \Pi_n}$, where $[A, b]$ concatenates column vector $b$ as a column of matrix $A$. Define $\boldsymbol{Z}^\star | \mathbf{A}, \boldsymbol{X}$ as the random variable with a uniform measure over the set $\Pi'(\mathbf{A}, \boldsymbol{X})$. We first prove that $\boldsymbol{Z}^\star | \mathbf{A}, \boldsymbol{X}$ is a most-expressive node embedding. Clearly, $\boldsymbol{Z}^\star | \mathbf{A}, \boldsymbol{X}$ is a node embedding, since the uniform measure over $\Pi'(\mathbf{A}, \boldsymbol{X})$ is $\mathcal{G}$-equivariant. All that is left to show is that we can construct $\Gamma^\star$ of any-size subset $S \in \mathcal{S}$ from $\boldsymbol{Z}^\star | \mathbf{A}, \boldsymbol{X}$ via

$$\Gamma^\star(\vec{S}, \mathbf{A}, \boldsymbol{X}) = \mathbb{E}_{\boldsymbol{Z}^\star}[f^{(|S|)}((\boldsymbol{Z}_v^\star)_{v \in S}) | \mathbf{A}, \boldsymbol{X}],$$

for some function $f^{(|S|)}$. This part of the proof has a constructive argument and comes in two parts.

Assume $S \in \mathcal{S}$ has no other joint isomorphic set of nodes in $\mathcal{S}$, i.e., $\nexists S_2 \in \mathcal{S}$ such that $S$ and $S_2$ are joint isomorphic in $(\mathbf{A}, \boldsymbol{X})$. For any such subset of nodes $S \in \mathcal{S}$, and any element $R_\pi \in \Pi'(\mathbf{A}, \boldsymbol{X})$, there is a bijective measurable map between the nodes in $S$ and their positions in the representation vector $R_\pi = (\Gamma^\star(v, \mathbf{A}, [\boldsymbol{X}, \pi(1, \ldots, n)^T]))_{\forall v \in V}$, since all node features are unique and there are no isomorphic nodes under such conditions. Consider the multiset

$$\mathcal{O}_S(\mathbf{A}, \boldsymbol{X}) := \{(\Gamma^\star(v, \mathbf{A}, [\boldsymbol{X}, \pi(1, \ldots, n)^T]))_{\forall v \in S}\}_{\forall \pi \in \Pi_n}$$

of the representations restricted to the set $S$. We now show that there exists an surjection between $\mathcal{O}_S(\mathbf{A}, \boldsymbol{X})$ and $\Gamma^\star(\vec{S}, \mathbf{A}, \boldsymbol{X})$. There is a surjection if for all $S_1, S_2 \in \mathcal{P}^\star(V)$ that are non-isomorphic, it implies $\mathcal{O}_{S_1}(\mathbf{A}, \boldsymbol{X}) \neq \mathcal{O}_{S_2}(\mathbf{A}, \boldsymbol{X})$. The condition is trivial if $|S_1| \neq |S_2|$ as $|\mathcal{O}_{S_1}(\mathbf{A}, \boldsymbol{X})| \neq |\mathcal{O}_{S_2}(\mathbf{A}, \boldsymbol{X})|$. If $|S_1| = |S_2|$, we prove by contradiction. Assume $\mathcal{O}_{S_1}(\mathbf{A}, \boldsymbol{X}) = \mathcal{O}_{S_2}(\mathbf{A}, \boldsymbol{X})$. Because of the unique feature ids and because $\Gamma^\star$ is most-expressive, the representation $\Gamma^\star(v, \mathbf{A}, [\boldsymbol{X}, \pi(1, \ldots, n)^T])$ of node $v \in V$ and permutation $\pi \in \Pi_n$ is unique. As $S_1$ is not isomorphic to $S_2$, and both sets have the same size, there must be at least one node $u \in S_1$ that has no isomorphic equivalent in $S_2$. Hence, there exists $\pi \in \Pi_n$ that gives a unique representation $\Gamma^\star(u, \mathbf{A}, [\boldsymbol{X}, \pi(1, \ldots, n)^T])$ that does not have a matching $\Gamma^\star(v, \mathbf{A}, [\boldsymbol{X}, \pi(1, \ldots, n)^T])$ for any $v \in S_2$ and $\pi' \in \Pi_n$. Therefore, $\exists a \in \mathcal{O}_{S_1}(\mathbf{A}, \boldsymbol{X})$, where $a \notin \mathcal{O}_{S_2}(\mathbf{A}, \boldsymbol{X})$, which is a contradiction since we assumed $\mathcal{O}_{S_1}(\mathbf{A}, \boldsymbol{X}) = \mathcal{O}_{S_2}(\mathbf{A}, \boldsymbol{X})$.

Now that we know there is such a surjection, a possible surjective measurable map between $\mathcal{O}_S(\mathbf{A}, \boldsymbol{X})$ and $\Gamma^\star(\vec{S}, \mathbf{A}, \boldsymbol{X})$ is a multiset function that takes $\mathcal{O}_S(\mathbf{A}, \boldsymbol{X})$ and outputs $\Gamma^\star(\vec{S}, \mathbf{A}, \boldsymbol{X})$. For finite multisets whose elements are real numbers $\mathbb{R}$, Wagstaff et al. (2019) shows that a most-expressive multiset function can be defined as the average of a function $f^{(|S|)}$ over the multiset. The elements of $\mathcal{O}_S(\mathbf{A}, \boldsymbol{X})$ are finite ordered sequences (ordered according to the permutation) and, thus, can be uniquely (bijectively) mapped to the real line with a measurable map, even when $\mathbf{A}$ and $\boldsymbol{X}$ have edge and node attributes defined over the real numbers $\mathbb{R}$. Thus, by Wagstaff et al. (2019), there exists some surjective function $f^{(|S|)}$ whose average over $\mathcal{O}_S(\mathbf{A}, \boldsymbol{X})$ give $\Gamma^\star(\vec{S}, \mathbf{A}, \boldsymbol{X})$.

Now assume $S_1, S_2 \subseteq V$ are joint isomorphic in $(\mathbf{A}, \boldsymbol{X})$, $S_1, S_2 \neq \emptyset$. Then, we have concluded that $\mathcal{O}_{S_1}(\mathbf{A}, \boldsymbol{X}) = \mathcal{O}_{S_2}(\mathbf{A}, \boldsymbol{X})$. Fortunately, by Definition 10, this non-uniqueness is a required property of the structural representations of $\Gamma^\star(S_1, \mathbf{A}, \boldsymbol{X})$ and $\Gamma^\star(S_2, \mathbf{A}, \boldsymbol{X})$, which must satisfy $\Gamma^\star(S_1, \mathbf{A}, \boldsymbol{X}) = \Gamma^\star(S_2, \mathbf{A}, \boldsymbol{X})$ if $S_1$ and $S_2$ are joint isomorphic, which concludes our proof. $\square$

Next, we restate and prove **Corollary 1**

**Corollary 1.** *The node embeddings in Definition 12 encompass embeddings given by matrix and tensor factorization methods —such as Singular Value Decomposition (SVD), Non-negative Matrix Factorization (NMF), implicit matrix factorization (a.k.a. word2vec)–, latent embeddings given by Bayesian graph models —such as Probabilistic Matrix Factorizations (PMFs) and variants—, variational autoencoder methods and graph neural networks that use random lighthouses to extract node embedddings.*

*Proof.* In Probabilistic Matrix Factorization (Mnih & Salakhutdinov, 2008), we have $\mathbf{A}_{uv} \sim \mathcal{N}(Z_u^T Z_v, \sigma_a^2 \boldsymbol{I})$ where $Z_u \sim \mathcal{N}(0, \sigma^2 \boldsymbol{I})$, $Z_v \sim \mathcal{N}(0, \sigma^2 \boldsymbol{I})$. We note that the posterior of $P(\boldsymbol{Z} | A)$ is clearly equivariant, satisfying definition 12, as a permutation action on the nodes requires the same permutation on the $\sigma^2 \boldsymbol{I}$ matrix as well to obtain $\boldsymbol{Z}$. The proof for Poisson Matrix Factorization (Gopalan et al., 2014a;b) follows a similar construction to the above, wherein the Normal assumption is replaced by the Poisson distribution.

Moreover, any matrix factorization algorithm gives an equivariant distribution of embeddings if the input matrices are randomly permuted upon input. Specifically, any Singular Value Decomposition (SVD) method satisfies Definition 12 as the distribution of the eigenvector solutions to degenerate singular values —which are invariant to unitary rotations in the corresponding degenerate eigenspace— will trivially be $\mathcal{G}$-equivariant even if the algorithm itself outputs values dependent on the node ids. Same is true for non-negative matrix factorization (Lee & Seung, 2001) and implicit matrix factorization (Levy & Goldberg, 2014; Mikolov et al., 2013).

PGNN's (You et al., 2019) compute the shortest distances between every node of the graph with a predetermined set of 'anchor' nodes to encode a distance metric. By definition, using such a distance metric would make the node embeddings learned by this technique $\mathcal{G}$-equivariant. The shortest path between all pairs of nodes in a graph can be seen equivalently as a function of a polynomial in $\mathbf{A}^k$. Alternatively, this can also be represented using the adjacency matrix and computed using the Floyd-Warshall algorithm (Cormen et al., 2009). The shortest distance is thus a function of $\mathbf{A}$ ignoring the node features $X$. Since the inputs to the GNN comprises of the distance metric, $\mathbf{A}$ and $X$, the node embeddings $Z$ can equivalently seen as a function of $\mathbf{A}$, $X$ and noise. The noise in this case is characterized by the randomized anchor set selection.

In variational auto-encoder models such as CVAE's, GVAE's, Graphite (Tang et al., 2019; Kipf & Welling, 2016b; Grover et al., 2019) the latent representations $Z$ are learned either via a mean field approximation or are sampled independently of each other i.e. $Z \sim P(\cdot | \mathbf{A}, X)$. We note that in the case of the mean field approximation, the probability distribution is a Dirac Delta. It is clear to see that the $Z$ learned in this case is $\mathcal{G}$-equivariant with respect to any permutation action of the nodes in the graph.

$\square$

Next, we restate and prove **Corollary 2**

**Corollary 2.** *The link prediction task between any two nodes $u, v \in V$ depends only on the most-expressive tuple representation $\Gamma^{\star}((u, v), \mathbf{A}, \mathbf{X})$. Moreover, $\Gamma^{\star}((u, v), \mathbf{A}, \mathbf{X})$ always exists for any graph $(\mathbf{A}, X)$ and nodes $(u, v)$. Finally, given most-expressive node embeddings $Z^{\star}$, there exists a function $f$ such that $\Gamma^{\star}((u, v), \mathbf{A}, \mathbf{X}) = \mathbb{E}_{Z^{\star}}[f(Z_u^{\star}, Z_v^{\star})], \forall u, v$.*

*Proof.* It is a consequence of Corollary 3 with $|S| = 2$. $\square$

Next, we restate and prove **Corollary 3**

**Corollary 3.** *Sample $Z$ according to Definition 12. Then, we can learn a $k$-node structural representation of a subset of $k$ nodes $S \subseteq V$, $|S| = k$, simply by learning a function $f^{(k)}$ whose average $\Gamma(\vec{S}, \mathbf{A}, X) = \mathbb{E}[f^{(k)}((Z_v)_{v \in S})]$ can be used to accurately predict $Y(\vec{S}, \mathbf{A}, X)$.*

*Proof.* This proof is a direct application of Theorem 2 which shows the statistical equivalence between node embeddings and strucutral representations.

Note that $f^{(k)}((Z_v)_{v \in S})$ can equivalently be represented as $f^{(k)}(\varphi(\Gamma(v, \mathbf{A}, X)_{v \in S}, \epsilon_S))$ using Theorem 2 and that the noise $\epsilon_S$ is marginalized from the noise distribution of Theorem 1, still preserving equivariance. With an assumption of the most powerful $f'^{(k)}$, which is able to capture dependencies within the node set (Murphy et al., 2018) and noise $\epsilon_S$, we can replace the above with $f'^{(k)}(\varphi(\Gamma(S, \mathbf{A}, X), \epsilon_S))$ and subsequently compute an expectation over this function to eliminate the noise.

$\square$

Next, we restate and prove **Corollary 4**

**Corollary 4.** *Transductive and inductive learning are unrelated to the concepts of node embeddings and structural representations.*

*Proof.* By Theorem 2, we can build most-expressive any-size joint representations from node embeddings, and we can get node embeddings from any-size most-expressive joint representations.

Hence, given enough computational resources, node embeddings and graph representations can have the same generalization performance over any tasks. This shows they are unrelated with the concepts of transductive and inductive learning. □

Next, we restate and prove **Corollary 5**

**Corollary 5.** *A node embeddings sampling scheme can increase the structural representation power of GNNs.*

*Proof.* The proof follows as a direct consequence of Theorem 2, along with Murphy et al. (2019) which demonstrates RP-GNN as a concrete method to do so. More specifically, appending unique node ids to node features uniformly at random, makes the nodes unique, and can be seen as a strategy to obtain node embeddings which satisfy Definition 12 using GNN's. By averaging over multiple such node embeddings gives us structural representations more powerful than that of standalone GNN's.

□

## 10 Colliding Graph Neural Networks (CGNNs)

In this section we propose a new variational auto-encoder procedure to obtain node embeddings using neural networks, denoted Colliding Graph Neural Networks (CGNNs). Our sole reason to propose a new auto-encoder method is because we want to test the expressiveness of node embedding auto-encoders —and, unfortunately, existing auto-encoders, such as Grover et al. (2019), do not properly account for the dependencies introduced by the colliding variables in the graphical model. In our experiments, shown later, we aggregate multiple node embedding sampled from CGNN to obtain structural representations of the corresponding nodes and node sets.

**Node Embedding Auto-encoder.** In CGNN's, we adopt a latent variable approach to learn node embeddings. Corresponding to each evidence feature vector $\boldsymbol{X}_{i,\cdot} \in \mathbb{R}^k \; \forall \; i \in V$, we introduce a latent variable $\boldsymbol{Z}_{i,\cdot} \in \mathbb{R}^k$. In addition, our graphical model also consists of observed variables $\mathbf{A}_{i,j,\cdot} \in \mathbb{R}^d \; \forall \; i,j \in V$. These are related through the joint distribution $p(\mathbf{A}, \boldsymbol{X}|\boldsymbol{Z}) = \prod_{i,j \in V \times V} p(\mathbf{A}_{i,j,\cdot}|\boldsymbol{Z}_{i,\cdot}, \boldsymbol{Z}_{j,\cdot}) \prod_{h \in V} p(\boldsymbol{X}_{h,\cdot}|\boldsymbol{Z}_{h,\cdot})$, which is summarized by the Bayesian network in Figure 5 in the Appendix. Note that $\mathbf{A}_{i,j,\cdot}$ is a collider, since it is observed and influenced by two hidden variables, $\boldsymbol{Z}_{i,\cdot}, \boldsymbol{Z}_{j,\cdot}$. A neural network is used to learn the joint probability via MCMC, in an unsupervised fashion. The model learns the parameters of the MCMC transition kernel via an unrolled Gibbs Sampler, a templated recurrent model (an MLP with shared weights across Gibbs sampling steps), partially inspired by Fan & Huang (2017).

The unrolled Gibbs Sampler, starts with a normal distribution of the latent variables $\boldsymbol{Z}_{i,\cdot}^{(0)}, \forall i \in V$, with each $\boldsymbol{Z}_{i,\cdot}^{(0)} \sim \mathcal{N}(\mathbf{0}, \mathbf{I})$ independently, where $I$ is the identity matrix. Subsequently at time steps $t = 1, 2, \ldots$, in accordance with the graphical model, each variable $\boldsymbol{Z}_{i,\cdot}^{(t)}$ is sequentially sampled from its true distribution conditioned on all observed edges of its corresponding node $i$, in addition to the most-up-to-date latent variables $\boldsymbol{Z}$'s associated with its immediate neighbors. The reparametrization trick (Kingma & Welling, 2013) allows us to backpropogate through the unrolled Gibbs Sampler. Algorithm 1 in the Appendix details our method. Consequently, this procedure has an effect on the run-time of this technique, which we alleviate by performing Parallel Gibbs Sampling by constructing parallel splashes (Gonzalez et al., 2011). Our unsupervised objective is reconstructing the noisy adjacency matrix.

## 11 CGNN Algorithms

The procedure to generate node embeddings used by the CGNN is given by Algorithm 1. Structural representations are computed as an unbiased estimate of the expected value of a function of the node embedding samples as given by Algorithm 2 via a set of sets function (Meng et al., 2019).

**input** : $\mathbf{A}$, $\boldsymbol{X}$, *num-times*
**output:** $\boldsymbol{Z}$
initialization: $\boldsymbol{Z}_u \sim \mathcal{N}(0, 1) \; \forall \; u \in V$
**while** *num-times > 0* **do**
    **for** $u \in V$ **do**
        $\forall v \in V$ such that $\mathbf{A}_{uv} = 1$
        hidden $\leftarrow f(\{Z_v\})$; // $f$ is a permutation invariant function
        visible $\leftarrow g(\{X_v\})$; // $g$ is a permutation invariant function
        $Z_u \leftarrow$ **MLP** (hidden, visible, $X_u$) **+ Noise** // With Reparametrization Trick
        // Equivalently, $\boldsymbol{Z_u} \sim P(\cdot | \{Z_v\}, \{X_v\}, \{A_{uv}\}, X_u)$
    **end**
    num-times $\leftarrow$ num-times - 1
**end**

**Algorithm 1:** Node Embeddings from the Unrolled Gibbs Sampler

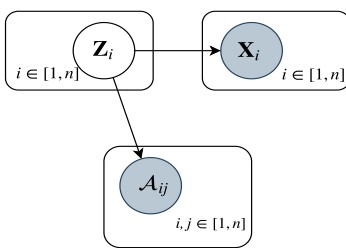

Figure 5: Latent variable model for Colliding Neural Networks . Observed evidence variables in gray

## 12   FURTHER RESULTS

In Table 2 we provide the results on node classification, link prediction and triad prediction on the Citeseer dataset.

## 13   DESCRIPTION OF DATASETS AND EXPERIMENTAL SETUP

A detailed description of the datasets and the splits are given in Table 3. Our implementation is in PyTorch using Python 3.6. The implementations for GIN and RP-GIN are done using the PyTorch Geometric Framework. We used two convolutional layers for GIN, RP-GIN since it had the best performance in our tasks (we had tested with 2/3/4/5 convolutional layers). Also since we perform tasks based on node representations rather than graph representations, we ignore the graph wide readout. For GIN and RP-GIN, the embedding dimension was set to 256 at both convolutional layers. All MLPS, across all models have 256 neurons. Optimization is performed with the Adam Optimizer (Kingma & Ba, 2014). For the GIN, RP-GIN the learning rate was tuned in {0.01, 0.001, 0.0001, 0.00001} whereas for CGNN's the learning rate was set to 0.001. Training was performed on Titan V GPU's. For more details refer to the code provided.

**input** : $\{\mathbf{Z}^{(i)}\}_{i=1}^{m}$, $k$ ;// node embedding samples, node set size
**output:** $g(\{\mathbf{Z}\}_{\mathcal{S}})$; $\mathcal{S} = \{S_1\}_{\forall S_1 \subset V : |S_1| = k}$ //structural representations, $\mathcal{S}$ is a set of sets
initialization: $g(\{\mathbf{Z}\}_{\mathcal{S}}) = \{\vec{0}\}$
**for** $i \in [1, m]$ **do**
   **for** $S \in \mathcal{S}$ **do**
      $g(\{Z_u\}_{u \in S}) \leftarrow g(\{Z_u\}_{u \in S}) + \frac{1}{m} f(\{Z_u^{(i)}\}_{u \in S})$ // $f$ is a permutation invariant function
   **end**
**end**

**Algorithm 2:** Structural Representations from the Node Embedding Samples

Table 2: Micro F1 score on three distinct tasks over the Citeseer dataset, averaged over 12 runs with standard deviation in parenthesis. The number within the parenthesis beside the model name indicates the number of Monte Carlo samples used in the estimation of the structural representation. MC-SVD$^{\dagger}(1)$ denotes the SVD procedure run until convergence with one Monte Carlo sample for the representation. Bold values show maximum empirical average, and multiple bolds happen when its standard deviation overlaps with another average.

|  | Node Classification | Link Prediction | Triad Prediction |
|---|---|---|---|
| *Random* | 0.167 | 0.500 | 0.250 |
| GIN(1) | 0.701(0.038) | 0.543(0.024) | 0.309(0.009) |
| GIN(5) | 0.706(0.044) | 0.525(0.040) | 0.311(0.022) |
| GIN(20) | 0.718(0.034) | 0.530(0.023) | 0.306(0.012) |
| RP-GIN(1) | 0.719(0.031) | 0.541(0.034) | 0.313(0.005) |
| RP-GIN(5) | 0.703(0.026) | 0.539(0.025) | 0.307(0.013) |
| RP-GIN(20) | 0.724(0.020) | 0.551(0.030) | 0.307(0.017) |
| 1-2-3 GNN(1) | 0.189(0.026) | 0.499(0.002) | 0.306(0.010) |
| 1-2-3 GNN(5) | 0.196(0.042) | 0.506(0.018) | 0.310(0.012) |
| 1-2-3 GNN(20) | 0.192(0.029) | 0.502(0.014) | 0.310(0.020) |
| MC-SVD$^{\dagger}$(1) | **0.733(0.007)** | 0.552(0.021) | 0.304(0.011) |
| MC-SVD(1) | **0.734(0.007)** | 0.562(0.017) | 0.297(0.015) |
| MC-SVD(5) | **0.739(0.006)** | 0.556(0.022) | 0.302(0.009) |
| MC-SVD(20) | **0.737(0.005)** | 0.565(0.020) | 0.299(0.015) |
| CGNN(1) | 0.689(0.010) | 0.598(0.024) | 0.305(0.009) |
| CGNN(5) | 0.713(0.009) | 0.627(0.048) | 0.301(0.013) |
| CGNN(20) | 0.721(0.008) | **0.654(0.049)** | 0.296(0.008) |

Table 3: Summary of the datasets

| CHARACTERISTIC | CORA | CITESEER | PUBMED | PPI |
|---|---|---|---|---|
| Number of Vertices | 2708 | 3327 | 19717 | 56944, 2373[a] |
| Number of Edges | 10556 | 9104 | 88648 | 819994, 41000[a] |
| Number of Vertex Features | 1433 | 3703 | 500 | 50 |
| Number of Classes | 7 | 6 | 3 | 121[b] |
| Number of Training Vertices | 1208 | 1827 | 18217 | 44906[c] |
| Number of Validation Vertices | 500 | 500 | 500 | 6514[c] |
| Number of Test Vertices | 1000 | 1000 | 1000 | 5524[c] |

[a] The PPI dataset comprises several graphs, so the quantities marked with an "a", represent the average characteristic of all graphs.
[b] For PPI, there are 121 targets, each taking values in $\{0, 1\}$.
[c] All of the training nodes come from 20 graphs while the validation and test nodes come from two graphs each not utilized during training.

