# OpenReview forum: "On the Equivalence between Positional Node Embeddings and Structural Graph Representations"
_ICLR.cc/2020/Conference — Accept (Poster)_

### Official Review · AnonReviewer2 · 2019-10-25
**Official Blind Review #2**

**Rating:** 8

**Review:**

The authors present mostly theoretical analysis indicating the equivalence of embeddings and structural graph representations. The authors argue that while most of the earlier work consider these to be different, they are actually the same and give theory and empirical results to back up this claim.

This is not an easy paper to read, as the authors immediately jump into heavy notation without much intuition or visual aid. It would be much better to include some figures to help the readers appreciate the work.
This continues throughout the experiments as well, where the authors are not very gentle when it comes to presentation.

I gave a weak accept as I do not want my (unfortunately) weak review (due to the paper topic not being my strong point) to have a great effect on the final decision. However, it is clear that the paper can and should be better written, and the paper ideas made closer to the readers. At the moment this is definitely not the case.


======= AFTER THE REBUTTAL ===========

Thank you for working on making the paper more accessible. I am perfectly fine with seeing this paper at the conference, and will change the vote to Accept simply to not block on my vote and to ensure the SPC makes sure to thoroughly consider this paper (in case they do some sort of ranking of paper reviews for deciding who gets accepted). However please note that my confidence remains as low as previously.
It is unfortunate that ICLR did not do any paper-reviewer matching like other conferences and that you are stuck with my weak review. But that is a different story that the organizers should make sure to address asap.

**Experience Assessment:**

I do not know much about this area.

**Review Assessment: Checking Correctness Of Derivations And Theory:**

I did not assess the derivations or theory.

**Review Assessment: Checking Correctness Of Experiments:**

I assessed the sensibility of the experiments.

**Review Assessment: Thoroughness In Paper Reading:**

I made a quick assessment of this paper.

---

> ### Author Response · Authors · 2019-11-09
> **Response to Official Blind Review #2**
>
> Thank you for the comments.
> Tackling this open problem, which brings matrix factorization (and other node embedding techniques) and graph neural networks under the same umbrella, redefines node embeddings (a 115 year-old concept), and fixing a decade of misunderstandings, requires the mathematical machinery of abstract algebra and modern probability theory, i.e., group and measure theories, respectively. We now connect some of the measure theory tools to causality, to help readers familiar with causal models. We found that balancing rigour, intuition, experiments, and page limit was extremely challenging for this paper. In a way, there may not be a perfect combination.
>
> However, we strive to make the paper an easier read to a general audience, since we believe this to be a fundamental contribution that will survive the test of time. In this regard, we have now added a new section in the appendix with a visual aid, it is worth checking. The added illustrations and discussion further showcases the kind of representations/ embeddings which these techniques learn, while emphasizing why the prevalent understanding of inductive and transductive learning is a misconception.
>
> We hope with our clarifications, the reviewer will see our fundamental contributions, and change to “Accept” (a fairer score given the significance of the work).

---

> > ### Author Response · Authors · 2019-11-15
> > **Any further questions or comments?**
> >
> > We believe we have addressed all your concerns. Please let us know if you have any further questions or comments.

---

### Official Review · AnonReviewer3 · 2019-10-29
**Official Blind Review #3**

**Rating:** 8

**Review:**

This paper proposes a unifield theoretical framework for node embeddings and structural graph representations, which bridgs methods like matrix factorization and graph neural networks.
The theoretical analysis is sufficient and experimental results are good. The theory alo shown that the concept of transductive and inductive learning is unrelated to node embeddings and graph representations, which clears another source of confusion in the literature.
In my opinion, this is a theory paper and the proof are sufficient.

**Experience Assessment:**

I have published one or two papers in this area.

**Review Assessment: Checking Correctness Of Derivations And Theory:**

I assessed the sensibility of the derivations and theory.

**Review Assessment: Checking Correctness Of Experiments:**

I assessed the sensibility of the experiments.

**Review Assessment: Thoroughness In Paper Reading:**

I read the paper at least twice and used my best judgement in assessing the paper.

---

> ### Author Response · Authors · 2019-11-09
> **Response to Official Blind Review #3**
>
> Thank you for the positive comments. And we would like to stress your importance in the discussion phase, as the only reviewer who has published papers in the area (the other two reviewers selected “I do not know much about this area”).
>
> We would like to emphasize that our work also has tremendous practical implications. Through it, we now understand why standard GNNs fail to predict links, and why methods that fix the issue (RGNN, SEAL, …) will propose randomized methods (because they rely on node embeddings to learn a joint two-node representation). Moreover, all these years we have been using Monte Carlo methods (matrix factorization, variational GNNs) without treating them as Monte Carlo: e.g., multiple matrix factorizations (MF’s) should be better than just one, and non-unique factorizations (variance), previously perceived as a foe in numerical methods, is now a potential friend. More precisely, we prove any graph task *must rely* on transforming the Monte Carlo samples (node embeddings) back into a structural representation. More samples => better representations. Hence, a method with more samples (where high-variance samples are not very informative) [e.g., MC-SVD] may be more accurate than a method with fewer samples (where samples are more informative) [e.g., SVD]. To the best of our knowledge, the research community was unaware of this fundamental trade-off.
>
> In addition, we also prove that all tasks that can be performed by node embeddings can also be performed by structural representations and vice-versa - which allow us to do tasks with GNNs and matrix factorization that were perceived to be beyond the reach of these methods, respectively. We also expect our results will spark a number of hybrid MF-GNN papers.

---

> > ### Author Response · Authors · 2019-11-15
> > **Any further questions or comments?**
> >
> > Please let us know if you have any further questions or comments.

---

### Official Review · AnonReviewer4 · 2019-11-03
**Official Blind Review #4**

**Rating:** 6

**Review:**

This paper tries to clarify some confounding concepts around graph/node representation learning by providing a unifying theoretical framework based on invariant theory. The authors define node embedding and structural graph representation based on a minimal number of requirements, which is, in turn, used to derive some useful properties and unify two seemly different concepts. Based on these, a new graph neural network framework is proposed. The experiments on multiple tasks with multiple datasets validate the main claim that a single node embedding is insufficient to capture the structural representation of a graph.

Overall, this paper tries to suggest a solution to a somewhat confusing and controversial problem that everyone recognises but hard to figure out a clear resolution. In that sense, this paper would serve as a good starting point by providing some theoretical baselines. I would like to see more discussion on this topic in future.

In many graph problems where each node is endowed with a certain feature set, we often observe the case where two subsets of nodes are isomorphic in terms of link structure but with different edge of node features. It would be good if there are some discussions on these cases.

Here's a minor comment on the representation of the paper. Although the concept of node embedding and structural representations get clearer as reading, the introduction, where I couldn't find any reference of these, seems unlikely to clarify the difference between these two concepts. The familiar representation in Section 3 might be the first part where the readers could get some intuition about their differences.



**Experience Assessment:**

I do not know much about this area.

**Review Assessment: Checking Correctness Of Derivations And Theory:**

I assessed the sensibility of the derivations and theory.

**Review Assessment: Checking Correctness Of Experiments:**

I assessed the sensibility of the experiments.

**Review Assessment: Thoroughness In Paper Reading:**

I read the paper at least twice and used my best judgement in assessing the paper.

---

> ### Author Response · Authors · 2019-11-09
> **Response to Official Blind Review #4**
>
> Thank you for the comments. Indeed, we also believe our work is foundational and will inspire new hybrid (matrix factorization - graph neural network) applied research in the future, besides more theoretical work. As you point out - our theory and results show that a single node embedding sample might be insufficient, and to this end we introduce new practical guidelines to  generating and using node embeddings (in the form of MC-SVD and CGNN’s), empowering them to be used for tasks which were previously perceived to be beyond the reach of factorization methods.
>
> Regarding the question about graphs with node/ edge features - the datasets we consider in our quantitative experiments are endowed with node features, whereas, to simplify the exposition, those in the qualitative experiments do not possess node or edge features. Our theory is designed to hold for any type of node and edge attributes (the empirical results showcases with node attributes).
>
> Introduction: We have now added a pointer to a new section in the appendix with a visual aid to help explain the concepts of structural representations and positional node embeddings. The added illustrations and discussion further showcases the kind of representations/ embeddings which these techniques learn, while emphasizing why the prevalent understanding of inductive and transductive learning is a misconception. Overall, we found that balancing rigour, intuition, experiments, and page limit was extremely challenging for this paper. In a way, there may not be a perfect combination. However, we strive to make the paper an easier read to a general audience and we welcome suggestions, since we believe this to be a fundamental contribution that will survive the test of time.
>
> We hope with our clarifications, the reviewer will see our fundamental contributions, and change to “Accept” (a fairer score given the significance of the work).

---

> > ### Author Response · Authors · 2019-11-15
> > **Any further questions or comments?**
> >
> > We believe we have addressed all your concerns. Please let us know if you have any further questions or comments.

---

### Author Response · Authors · 2019-11-09
**Paper Revised**

We have added a pointer in the introduction to a new section in the appendix that helps visually illustrate the concepts of structural representations and positional node embeddings. A new lemma (relating causality and noise outsourcing) has been added to help readers familiar with causality understand the implications of Theorem 2 (it also applies to causal models).

---

### Author Response · Authors · 2019-11-14
**Paper Revised**

On further reviewing the proof of Theorem 2, we have made the proof easier to follow by fixing a bug in first paragraph the proof. The condition is easier to understand since it clearly follows from Theorem 1. Moreover, in Theorem 1 we further emphasized that Y is defined over a "set" S (A set is invariant to its ordering). It was obvious since isomorphic sets must have the same distribution over Y, but we feel this may further help future readers.

---

### Public Comment · ~Ziwei_Zhang1 · 2020-01-08
**Regarding Experiments in Section 4.3**

Thank you for this insightful work!
However, I do have a question regarding "Structural node representations from node embeddings" in Section 4.3. The authors mention running multiple SVD (until convergence) "with the sources of randomness being due to a random permutation of the adjacency matrix given as input to the SVD method and the random seed it uses". However, from linear algebra we know the results of SVD are deterministic (up to a sign, which can be easily dealt with). For the food web example, the graph contains two set of eigenvectors (with the same eigenvalue), each set having all-zero elements for a connected component. As a result, the 'randomness' seems to be purely from how to arrange these eigenvectors? (In other words, for a graph with distinct eigenvalues, the results should be exactly the same). In any case, I would suggest a normal matrix factorization using gradient descent be a better example, since the randomness is clearer in MF than SVD.

---

> ### Author Response · Authors · 2020-01-09
> **Response to Question Regarding Experiments in Section 4.3**
>
> Thank you for your comment.
> With regard to your comment, we use multiple SVD samples (not run until convergence - just one step of optimization) for the results with just MC-SVD (This is also the case for the food web example). SVD is run until convergence only when we specify MC-SVD $^\dagger$  (with the dagger superscript) in Table 1. To address your other question, SVD run until convergence is unique only when the graph is devoid of isomorphic nodes, otherwise even this would require multiple samples.
> We will update the manuscript to make this more explict.
> Thanks,
> Authors

---

### Public Comment · ~Oh-Hyun_Kwon1 · 2020-01-27
**Where can I find the source code?**

Amazing work!
I am looking for the source code for this paper (esp. CGNN and MC-SVD).
The Appendix mentioned that the code is provided but I am not sure where I can find it.
Could you let me know where can I find it?
Thanks!

---

> ### Author Response · Authors · 2020-02-07
> **Reply to comment**
>
> Thank you very much for the comment and the interest in our work. We are  working to get a complete repository with all the models,  which are easily reusable in other settings as well.

---

### Author Response · Authors · 2020-09-22
**Paper Revised**

This version corrects some typos in the definition of $\Sigma$, it should be $\Sigma_n$
Arxiv Link -> https://arxiv.org/abs/1910.00452
Code -> https://github.com/PurdueMINDS/Equivalence

---

### Decision · Program_Chairs · 2019-12-19

**Decision:**

Accept (Poster)

**Comment:**

The paper shows the relationship between node embeddings and structural graph representations. By careful definition of what structural node representation means, and what node embedding means, using the permutation group, the authors show in Theorem 2 that node embeddings cannot represent any extra information that is not already in the structural representation. The paper then provide empirical experiments on three tasks, and show in a fourth task an illustration of the theoretical results.

The reviewers of the paper scored the paper highly, but with low confidence. I read the paper myself (unfortunately not with a lot of time), with the aim of increasing the confidence of the resulting decision. The main gap in the paper is between the phrases "structural node representation" and "node embedding", and their theoretical definitions. The analogy of distribution and its samples follows unsurprisingly from the definitions (8 and 12), but the interpretation of those definitions as the corresponding English phrases is not obvious by only looking at the definitions. There also seems to be a sleight of hand going on with the most expressive representations (Definitions 9 and 11), which is used to make the conditional independence statement of Theorem 2. The authors should clarify in the final version whether the existence of such a representation can be shown, or even better a constructive way to get it from data.

Given the significance of the theoretical results, the authors should improve the introduction of the two main concepts by:
- relating them to prior work (one way is to move Section 5 towards the front)
- explaining in greater detail why Definitions 8 and 12 correspond to the two concepts. For example expanding the part of the proof of Corollary 1 about SVD, to make clear what Definition 12 means.
- a corresponding simple example of Definition 8 to relate to a classical method.

The paper provides a nice connection between two disparate concepts. Unfortunately, the connection uses graph invariance and equivariance, which is unfamiliar to many of the ICLR audience. On balance, I believe that the authors can improve the presentation such that a reader can understand the implications of the connection without being an expert in graph isomorphism. As such, I am recommending an accept.